# TIGER: Text-Informed Generalized Enzyme-Reaction Retrieval

## Abstract

Enzyme–reaction retrieval is a fundamental problem in computational biology, underpinning enzyme characterization, reaction mechanism elucidation, and the rational design of metabolic pathways and biocatalysts. As a bidirectional task, it entails both enzyme-to-reaction and reaction-to-enzyme mapping. However, existing approaches suffer from poor generalization across tasks and distributions, with performance highly sensitive to dataset splits and substantial asymmetry between retrieval directions. To address these challenges, we present TIGER, a Text-Informed Generalized Enzyme-Reaction Retrieval framework that leverages protein-to-text generation models to distill textual semantic knowledge from enzyme sequences, providing a generalized representation that bridges enzymes and biochemical reactions. To ensure the quality and reliability of textual semantics, we design a Dynamic Gating Network that adaptively fuses text-derived knowledge with sequence features, enabling more consistent and informative enzyme representations, while a Structure-Shared Feature Projector aligns enzyme and reaction representations within a unified latent space. Extensive experiments demonstrate that, under bidirectional retrieval supervision, TIGER significantly outperforms state-of-the-art baselines across diverse distributions and exhibits strong robustness and transferability across tasks.

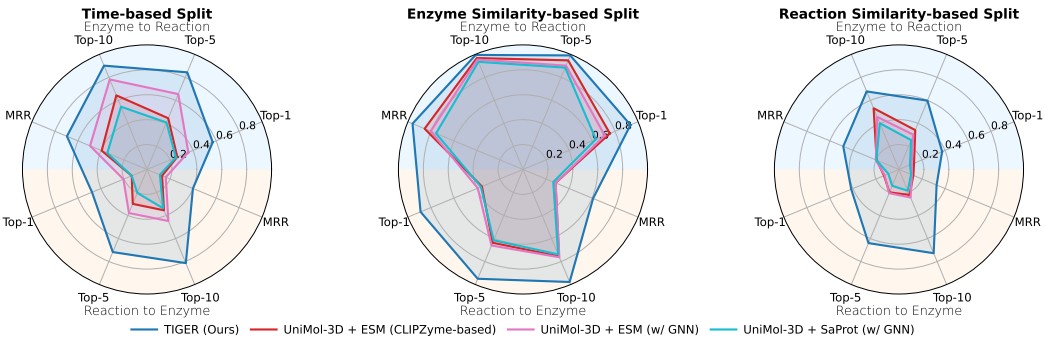

Figure 1: Retrieval performance of TIGER and existing methods under time-, enzyme similarity-, and reaction similarity-based splits on ReactZyme, demonstrating the robust generalization capacity of TIGER across heterogeneous evaluation settings.

## 1 Introduction

Enzymes (Buller et al., 2023; Benítez-Mateos et al., 2022), as the central class of biocatalysts, are pivotal in orchestrating biochemical transformations essential for life. While traditional bioinformatics has increasingly emphasized predictive tasks such as EC number classification (Yu et al., 2023; Dalkiran et al., 2018), the emerging field of Enzyme-Reaction Retrieval adopts a relational computational paradigm. By establishing bidirectional correspondences between enzymes and their catalyzed reactions, it enables systematic exploration of functional diversity, pathway reconstruction, and applications in synthetic biology. With the rapid growth of high-throughput sequencing (Pai &

Satpathy, 2021; Hoffman et al., 2014), the scale and complexity of enzyme–reaction data provide fertile ground for these advanced modeling paradigms.

Existing computational approaches for enzyme-reaction retrieval have predominantly relied on contrastive learning paradigms Mikhael et al. (2024), which align representations derived from enzyme sequences with those from chemical reactions. However, these frameworks exhibit notable limitations that hinder their practical application Hua et al. (2024). First, they demonstrate cross-directional asymmetry, where the retrieval accuracy from enzymes to reactions substantially diverges from the reverse direction. This asymmetry reveals a fundamental inconsistency in semantic alignment and a lack of representational coherence. Second, these models show a high sensitivity to dataset splits, with performance fluctuating significantly under different partition strategies. This instability points to a critical lack of generalization ability and raises concerns about their robustness across heterogeneous data distributions.

A key reason of the performance gap is that pre-trained protein models are not explicitly trained to fundamentally understand chemical transformations. Their pre-training tasks focus on structural and evolutionary information (Lin et al., 2023; Jumper et al., 2021; Wang et al., 2022), neglecting to learn the subtle, reaction-specific features required for catalysis. To bridge this gap, we draw inspiration from knowledge-enhanced multimodal retrieval paradigms, which have achieved remarkable success in fields like image-text matching (Mi et al., 2024; Feng et al., 2023; Suo et al., 2024). We propose a novel framework, Text-Informed Generalized Enzyme-Reaction Retrieval (TIGER), that leverages protein-to-text generation models (Liu et al., 2024; Abdine et al., 2024) to generate rich, textual, knowledge-rich representations of enzymes. TIGER moves beyond a purely sequential or structural understanding by producing descriptive summaries that explicitly incorporate an enzyme's catalytic function, substrate interactions, and other key details derived from its associated chemical reactions. Our approach treats the textual descriptions as a form of knowledge augmentation, enabling the model to create a more symmetric and semantically coherent joint embedding space.

As textual descriptions generated by pre-trained language models are prone to semantic noise (Cao et al., 2025; Liang et al., 2024) and "hallucinations" (Vishwanath et al., 2024; Jesson et al., 2024), we introduce a Dynamic Gating Network (DGN) to adaptively regulate their contribution during representation learning. Instead of treating all textual inputs equally, the DGN learns reliability-aware gating weights that reflect the semantic consistency of textual embeddings with enzyme sequence features. Reliable descriptions are thus emphasized to enrich biochemical semantics, while noisy or irrelevant ones are down-weighted to prevent spurious correlations. This adaptive modulation enables the framework to retain the complementary knowledge conveyed by textual cues while enhancing robustness against imperfect supervision, ultimately leading to more stable training and stronger generalization in enzyme–reaction retrieval.

To enhance representational coherence and cross-modal generalization, we introduce a Structure-Shared Feature Projector that maps enzyme and reaction embeddings into a unified latent space. We trained TIGER with bidirectional contrastive supervision and evaluated it on ReactZyme, the largest enzyme–reaction dataset available. As shown in Figure 1, TIGER consistently outperforms representative baselines across time-based, enzyme similarity-based, and reaction similarity-based splits, demonstrating superior retrieval accuracy and robustness. These results highlight its strong generalization ability under diverse conditions and underscore the effectiveness of the text-informed design. In summary, our main contributions are:

- We propose TIGER, a text-informed generalized enzyme–reaction retrieval framework that incorporates knowledge-rich descriptions to establish a more symmetric and semantically coherent embedding space.

- To address the potential errors and hallucinations in AI-Generated textual descriptions, we design a Dynamic Gating Network, which adaptively balances the contributions from sequences and texts to ensure reliable integration, thereby enhancing robustness and cross-modal generalization.

- We conduct comprehensive experiments on ReactZyme, where TIGER consistently achieves state-of-the-art performance, yielding relative Hit@1 improvements ranging from 14% to over 200% across diverse evaluation splits, with ablation studies confirming the contribution of each component.

## 2 RELATED WORK

**Enzyme-Reaction Retrieval** Traditional enzyme studies have largely focused on EC classification (Dalkiran et al., 2018; Yu et al., 2023) and substrate binding (Zeng et al., 2022), yet such categorical tasks (Fernstad & Johansson, 2011) overlook the richer relational structure between enzymes and the reactions they catalyze. Enzyme-reaction retrieval has thus emerged as a more flexible paradigm, learning a shared embedding space to directly align and rank enzymes with reactions. Early attempts such as CLIPZyme (Mikhael et al., 2024) leveraged contrastive learning to couple enzymatic sequences with reaction representations, offering an relatively initial yet advanced formulation of enzyme–reaction retrieval. To further advance this emerging direction, ReactZyme (Hua et al., 2024) established a large-scale standardized benchmark that integrates diverse enzyme–reaction data and enables systematic evaluation across multiple methodologies. In particular, it benchmarked a spectrum of baselines, including 2D and 3D molecular encoders for reactions (MAT-2D/3D (Maziarka et al., 2020), UniMol-2D/3D (Zhou et al., 2023)), protein language models for enzymes (ESM (Lin et al., 2023), SaProt (Su et al., 2023)), and residue-level equivariant graph networks (FANN) (Puny et al., 2021), thereby providing a comprehensive testbed for cross-modal retrieval. Importantly, these experiments further revealed open challenges, including cross-directional asymmetry and sensitivity to dataset splits, underscoring the need for more robust and semantically grounded retrieval frameworks.

**Text-informed Protein Representation Learning** Protein representation learning has historically centered on amino acid sequences, yet such unimodal formulations inherently neglect the rich functional and mechanistic semantics embedded in biomedical corpora. Recent advances have therefore shifted towards text-informed paradigms, wherein protein sequences are complemented with natural language supervision to derive more expressive and functionally grounded embeddings. Early work in this direction includes ProtST (Xu et al., 2023), which introduced a dual-modal contrastive framework aligning protein sequences with biomedical text, showing that textual context provides orthogonal signals to boost downstream performance. ProTrek (Su et al., 2024) extended this to a tri-modal setting by jointly modeling sequences, structures, and textual annotations, underscoring the complementary role of structural information. Inspired by CLIP Radford et al. (2021), approaches such as ProtCLIP (Zhou et al., 2025) and ProteinCLIP (Wu et al., 2024) further coupled large-scale protein language models with curated text, producing semantically coherent and function-aware representations. Beyond general-purpose embeddings, text-informed frameworks have proven effective in specialized bioinformatics tasks. MMSite (Ouyang et al., 2024) leveraged textual cues with sequence and structure for active-site identification, while BioT5 (Pei et al., 2023) introduced a cross-modal pre-training paradigm that integrates chemical knowledge with natural language. Similarly, CoSEF-DBP (Zhang et al., 2025) and ProteinDT (Liu et al., 2025) demonstrated the utility of text guidance in identifying DNA-binding proteins and protein design. Collectively, these studies highlight that textual knowledge provides critical signals for tasks ranging from function annotation and interaction analysis to generative protein engineering. In this work, we adopt this paradigm and introduce a textual quality control mechanism to address reliability issues in textual supervision, thereby enhancing performance and generalization in the enzyme–reaction retrieval task.

## 3 TASK FORMULATION

Let $\mathcal{E} = \{e_1, e_2, \ldots, e_N\}$ denote a set of enzyme entities and $\mathcal{R} = \{r_1, r_2, \ldots, r_M\}$ denote a set of biochemical reactions. Each enzyme $e_i \in \mathcal{E}$ may be associated with one or more reactions $r_j \in \mathcal{R}$ that it catalyzes or participates in, and vice versa.

We assume access to a partial ground-truth correspondence $\mathcal{A} \subseteq \mathcal{E} \times \mathcal{R}$, where each pair $(e, r) \in \mathcal{A}$ indicates a known biological association. The goal is to recover or rank such associations via representation-based matching in a learned metric space.

Formally, we aim to learn an embedding function

$$\phi_1 : \mathcal{E} \to \mathbb{R}^d, \phi_2 : \mathcal{R} \to \mathbb{R}^d$$

that maps both enzymes and reactions into a shared latent space $\mathbb{R}^d$ such that semantically associated pairs are close under a similarity metric $s : \mathbb{R}^d \times \mathbb{R}^d \to \mathbb{R}$. For a matched pair $(e, r) \in \mathcal{A}$, the embedding function should satisfy

$$s(\phi_1(e), \phi_2(r)) \gg s(\phi_1(e), \phi_2(r')) \quad \forall r' \in \mathcal{R}, (e, r') \notin \mathcal{A},$$

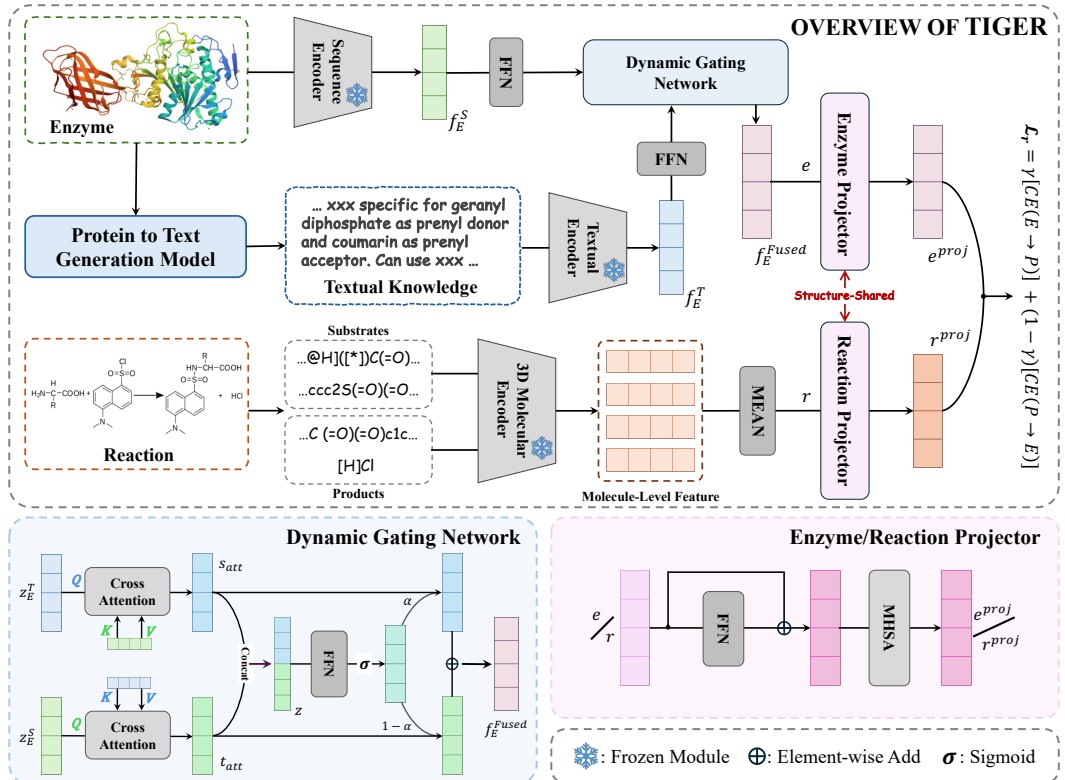

Figure 2: Overview of the proposed TIGER framework. Enzyme sequences and generated textual Knowledge are adaptively fused by a Dynamic Gating Network, while reactions are represented through a 3D molecular encoder. The two modalities are projected into a shared embedding space via Structure-Shared Feature Projectors and jointly optimized with bidirectional contrastive learning for generalized enzyme–reaction retrieval.

and symmetrically,

$$s(\phi_2(r), \phi_1(e)) \gg s(\phi_2(r), \phi_1(e')) \quad \forall e' \in \mathcal{E}, \, (e', r) \notin \mathcal{A}.$$

This problem setting naturally defines a bidirectional retrieval task:

- **Enzyme-to-Reaction (E2R)**: Given an enzyme query $e \in \mathcal{E}$, retrieve the most relevant reaction(s) $r \in \mathcal{R}$ such that $(e, r) \in \mathcal{A}$.

- **Reaction-to-Enzyme (R2E)**: Given a reaction query $r \in \mathcal{R}$, retrieve the most relevant enzyme(s) $e \in \mathcal{E}$ such that $(e, r) \in \mathcal{A}$.

This bidirectional matching formulation provides a foundation for downstream applications such as enzyme function prediction, metabolic pathway reconstruction, and biochemical knowledge graph completion. The subsequent sections describe our multimodal representation learning framework and the training procedure used to optimize $\phi_1$ and $\phi_2$ under contrastive supervision.

## 4 OUR APPROACH

TIGER is fundamentally designed under a contrastive learning paradigm. As shown in Figure 2, the framework mainly consists of two branches: multimodal enzyme representation learning and reaction representation learning, which are jointly optimized through bidirectional contrastive learning to capture their underlying semantic correspondence. On the enzyme side, sequence embeddings from a pre-trained protein language model are fused with textual semantics via a Dynamic Gating Network to balance complementary information and suppress noise. On the reaction side, 3D

molecular encoders extract structural representations of substrates and products, which are aggregated into reaction-level embeddings. Both branches are mapped into a unified embedding space through Structure-Shared Feature Projectors, ensuring symmetric alignment and robust generalization for retrieval.

## 4.1 MULTIMODAL ENZYME REPRESENTATION LEARNING

For any enzyme $e \in \mathcal{E}$, we denote its amino acid sequence as $s_e$. Based on $s_e$, we obtain an automatically generated textual knowledge $t_e$ through a protein-to-text generation model, specifically ESM2Text. To construct a robust enzyme representation, we fuse the sequence embedding and textual embedding using a Dynamic Gating Network, which adaptively balances complementary information while suppressing noise from generated text.

### 4.1.1 MULTIMODAL FEATURE EXTRACTING

For each enzyme $e \in \mathcal{E}$, we derive modality-specific representations from both sequence and textual views. The amino acid sequence $s_e$ is encoded by the pretrained protein language model ESM2 (Lin et al., 2023), which effectively captures contextualized dependencies along the primary structure:

$$f_E^S = \psi_{\text{seq}}(s_e),$$

while the automatically generated textual description $t_e$ is embedded using PubMedBERT (Gu et al., 2021), a domain-specific language model trained on large-scale biomedical literature:

$$f_E^T = \psi_{\text{text}}(t_e).$$

To obtain task-adaptive and dimensionally consistent representations, both embeddings are further transformed through modality-specific feed-forward networks:

$$z_e^S = \text{FFN}_S(f_E^S), \quad z_e^T = \text{FFN}_T(f_E^T).$$

In this way, we obtain two complementary features $z_e^S$ and $z_e^T$, where the former emphasizes structural and sequential context, and the latter conveys functional and semantic information. These features jointly form the foundation for unified enzyme embeddings.

### 4.1.2 DYNAMIC GATING NETWORK

The histogram in Figure 3 illustrates the cosine similarity distribution between AI-generated textual knowledge (via ESM2Text) and human-reviewed SwissProt (Bairoch & Apweiler, 2000; uni, 2025) descriptions, focusing on the subset with similarity below 0.95. This subset constitutes approximately one-third of the entire dataset, suggesting that while most generated

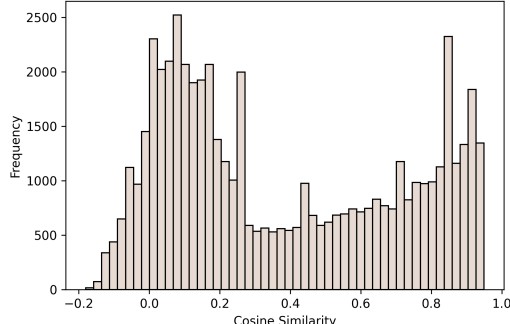

Figure 3: Cosine Similarities between AI-Generated and SwissProt (Human-reviewed) Textual Knowledge

texts remain consistent with curated references, a non-negligible portion exhibits notable semantic deviations. Such discrepancies introduce noise into cross-modal alignment and may undermine downstream performance, thereby highlighting the importance of text quality control. To this end, we propose the *Dynamic Gating Network*, a reliability-aware integration mechanism that adaptively modulates the contribution of textual features according to their estimated quality, ultimately improving the robustness of multimodal enzyme representations.

Building on this motivation, the Dynamic Gating Network operates on the features $z_e^S$ and $z_e^T$, progressively integrating them through cross-modal attention and adaptive gating. We first employ bidirectional multi-head attention to enable semantic refinement across modalities:

$$s_{att} = \text{MHA}(z_e^S, z_e^T, z_e^T), \quad t_{att} = \text{MHA}(z_e^T, z_e^S, z_e^S).$$

The attended features are then combined via a gating mechanism that estimates their relative reliability. Specifically, a joint representation is constructed as

$$z = [s_{att} \| t_{att}],$$

from which a gating coefficient is derived:

$$\boldsymbol{\alpha} = \sigma(W_g z),$$

where $\sigma$ denotes the sigmoid function. The gated fusion is computed as

$$f_{gated} = \boldsymbol{\alpha} \odot s_{att} + (1 - \boldsymbol{\alpha}) \odot t_{att}.$$

Finally, to stabilize the integration, we concatenate $f_{gated}$ with the aggregated signal $(s_{att} + t_{att})$ and apply a feed-forward transformation:

$$f_E^{Fused} = \text{FFN}_{fuse}\Big([f_{gated} \,\|\, (s_{att} + t_{att})]\Big).$$

The resulting representation $f_E^{Fused} \in \mathbb{R}^d$ serves as the unified enzyme representation, providing a robust and reliability-aware basis for contrastive learning against reaction representations.

## 4.2 REACTION REPRESENTATION LEARNING.

For each biochemical reaction $r \in \mathcal{R}$, we follow prior studies that have demonstrated the effectiveness of molecular pre-trained models in capturing reaction semantics, and adopt UniMol-3D (Maziarka et al., 2020), one of the most widely used and high-performing molecular encoders. Specifically, the reaction is decomposed into its constituent substrates and products. Each molecule is independently encoded by UniMol-3D, which leverages both graph-level and 3D conformational information to generate chemically meaningful representations. This strategy ensures that stereochemical and geometric cues, which are often critical for catalytic processes, are faithfully preserved. The resulting molecular embeddings are then aggregated to form the reaction-level representation. Concretely, we compute the reaction embedding by averaging over all encoded substrates and products:

$$\boldsymbol{r} = \frac{1}{|\mathcal{S}| + |\mathcal{P}|} \left( \sum_{s \in \mathcal{S}} \text{UniMol}(s) + \sum_{p \in \mathcal{P}} \text{UniMol}(p) \right), \tag{1}$$

where $\mathcal{S}$ and $\mathcal{P}$ denote the sets of substrates and products, respectively. This design choice, consistent with the standard practice in recent benchmark works, provides a simple yet robust way to derive reaction features that capture both local molecular structure and global reaction context.

The extracted reaction representation and the enzyme representation introduced are subsequently projected into a shared latent space via the Structural-Shared Feature Projector. This architecture enables cross-modal alignment under contrastive supervision and facilitates bidirectional retrieval between enzymes and reactions.

## 4.3 STRUCTURE-SHARED FEATURE PROJECTOR

To enable effective enzyme-reaction retrieval, we introduce the Structural-Shared Feature Projector, a dual-branch module that maps heterogeneous inputs into a unified embedding space. Each modality is transformed through a symmetric pipeline of non-linear encoding, residual connections, attention-based contextualization, and final projection:

$$\phi(\mathbf{x}) = \text{LN}_{proj}(\text{MHSA}(\text{FFN}(\mathbf{x}) + \mathbf{W}_{res}\mathbf{x})), \tag{2}$$

where $\mathbf{x} \in \{e, r\}$, $\text{LN}_{proj}$ ensures dimensional consistency, and $\mathbf{W}_{res}$ provides residual enhancement. The projected embeddings $\phi(e)$ and $\phi(r)$ lie in a shared space $\mathbb{R}^d$, with a learnable temperature $\tau$ scaling pairwise similarities during contrastive training.

This design enforces semantic proximity of enzyme-reaction pairs, thereby enabling robust bidirectional retrieval under contrastive supervision.

## 4.4 CONTRASTIVE TRAINING OBJECTIVE

We employ a symmetric contrastive learning objective to align enzyme and reaction representations in a shared latent space. Given a batch of $N$ enzyme–reaction pairs $\{(e_i, r_i)\}_{i=1}^N$, the cosine similarity between projected embeddings is defined as

$$s_{ij} = \frac{e_i^{proj} \cdot r_j^{proj}}{\tau \|e_i^{proj}\| \|r_j^{proj}\|},$$

where $\tau > 0$ is a learnable temperature parameter. The bidirectional losses are

$$\mathcal{L}_{e2r} = -\frac{1}{N}\sum_{i=1}^{N}\log\frac{\exp(s_{ii})}{\sum_{j=1}^{N}\exp(s_{ij})}, \quad \mathcal{L}_{r2e} = -\frac{1}{N}\sum_{i=1}^{N}\log\frac{\exp(s_{ii})}{\sum_{j=1}^{N}\exp(s_{ji})},$$

and the final retrieval loss is

$$\mathcal{L}_r = \gamma\mathcal{L}_{e2r} + (1-\gamma)\mathcal{L}_{r2e}.$$

where $\gamma$ balances two retrieval directions .

## 5 EXPERIMENTS AND ANALYSIS

### 5.1 DATASET: REACTZYME

**ReactZyme (Hua et al., 2024)** is the latest and most comprehensive benchmark for enzyme–reaction retrieval, constructed from curated SwissProt and Rhea resources. It contains over 178K enzyme–reaction associations, spanning more than 178K unique enzymes and 7.7K distinct reactions, thereby providing a functionally grounded alternative to traditional EC- or ontology-based annotations. To rigorously assess generalization, ReactZyme defines three complementary evaluation splits: **time-based**, where training pairs precede a temporal cutoff while later pairs are reserved for testing; **enzyme similarity–based**, where test enzymes are sequence-dissimilar to those in training; and **reaction similarity–based**, where test reactions are entirely absent from training. These settings form a progressive hierarchy of difficulty, with the reaction similarity split posing the greatest challenge as it requires extrapolation to unseen chemical transformations.

### 5.2 RESULTS ANALYSIS

Our evaluation on the ReactZyme dataset examines how TIGER improves retrieval performance and generalization compared to existing baselines. We further analyze how textual knowledge, together with mechanisms such as the Dynamic Gating Network and the loss setting, contributes to the overall effectiveness of the framework. The following results provide quantitative evidence for these improvements.

#### 5.2.1 PERFORMANCE ANALYSIS OF TIGER

Table 1 presents a comprehensive comparison between TIGER and representative baselines across three evaluation splits. For clarity, the superscripts denote the encoder configurations used by the baselines under the ReactZyme protocol: [1]UniMol-3D for reactions combined with ESM for enzymes, [2]MAT-2D with ESM, and [3]UniMol-3D with SaProt. These combinations are widely adopted in molecular representation learning and thus serve as strong reference settings for benchmarking.

From the quantitative results, three salient observations can be drawn. First, in terms of absolute accuracy, **TIGER consistently surpasses all baseline methods.** For example, under the time-based split, TIGER achieves a Hit@1 of 0.581 in the enzyme-to-reaction direction, compared to the strongest baseline (Bi-RNN[2]) at 0.391, representing a relative improvement of more than 48%. In the reverse direction, TIGER also achieves 0.454 Hit@1, substantially higher than the baseline maximum of 0.265. These gains demonstrate that TIGER remains highly effective under temporally disjoint training and test distributions, which approximate real-world deployment scenarios.

Second, **TIGER exhibits greater bidirectional consistency**. While prior methods often excel in one retrieval direction but perform poorly in the other (e.g., CLIPZyme[1] achieves 0.755 Hit@1 for enzyme-to-reaction yet only 0.357 for reaction-to-enzyme), TIGER maintains strong and relatively balanced results (0.931 vs. 0.792 in the enzyme similarity-based split). This reduction in directional asymmetry highlights the effectiveness of our modality-aware alignment in constructing semantically coherent cross-modal representations.

Third, **TIGER demonstrates robustness across heterogeneous evaluation conditions**. In the reaction similarity-based split—arguably the most challenging due to minimal structural overlap between training and test reactions—baselines generally suffer severe degradation (e.g., GNN[3] drops to 0.096 Hit@1 for enzyme-to-reaction). In contrast, TIGER secures 0.416 Hit@1 and 0.430 in

Table 1: Performance comparison across three splits on ReactZyme. The encoders used by the baselines: [1]UniMol-3D + ESM, [2]MAT-2D + ESM, [3]UniMol-3D + SaProt

| Split | Time-based Split | | | | Enzyme Similarity-based Split | | | | Reaction Similarity-based Split | | | |
|---|---|---|---|---|---|---|---|---|---|---|---|---|
| Direction | E→R | | R→E | | E→R | | R→E | | E→R | | R→E | |
| Method | Hit@1 | MRR | Hit@1 | MRR | Hit@1 | MRR | Hit@1 | MRR | Hit@1 | MRR | Hit@1 | MRR |
| ReactZyme[1] | 0.291 | 0.410 | 0.168 | 0.140 | 0.727 | 0.811 | 0.409 | 0.293 | 0.091 | 0.201 | 0.135 | 0.134 |
| ReactZyme[2] | 0.325 | 0.218 | 0.218 | 0.179 | 0.599 | 0.728 | 0.362 | 0.259 | 0.109 | 0.199 | 0.093 | 0.096 |
| ReactZyme[3] | 0.092 | 0.159 | 0.056 | 0.054 | 0.600 | 0.723 | 0.348 | 0.256 | 0.094 | 0.194 | 0.114 | 0.104 |
| Fingerprint | 0.236 | 0.298 | 0.144 | 0.117 | 0.579 | 0.639 | 0.255 | 0.204 | 0.094 | 0.194 | 0.114 | 0.104 |
| GNN[1] | 0.359 | 0.495 | 0.205 | 0.163 | 0.711 | 0.802 | 0.393 | 0.284 | 0.110 | 0.201 | 0.124 | 0.113 |
| GNN[3] | 0.251 | 0.345 | 0.133 | 0.112 | 0.633 | 0.746 | 0.366 | 0.263 | 0.096 | 0.197 | 0.092 | 0.105 |
| Bi-RNN[1] | 0.354 | 0.494 | 0.254 | 0.211 | 0.811 | 0.875 | 0.509 | 0.387 | 0.109 | 0.197 | 0.124 | 0.121 |
| Bi-RNN[2] | 0.391 | 0.530 | 0.265 | 0.227 | 0.815 | 0.886 | 0.589 | 0.456 | 0.118 | 0.240 | 0.171 | 0.170 |
| CLIPZyme[1] | 0.263 | 0.394 | 0.133 | 0.131 | 0.755 | 0.855 | 0.357 | 0.283 | 0.131 | 0.194 | 0.130 | 0.125 |
| CLIPZyme[2] | 0.304 | 0.436 | 0.176 | 0.168 | 0.549 | 0.697 | 0.334 | 0.204 | 0.124 | 0.220 | 0.146 | 0.152 |
| TIGER (Ours) | **0.581** | **0.690** | **0.454** | **0.366** | **0.931** | **0.956** | **0.792** | **0.592** | **0.416** | **0.518** | **0.430** | **0.319** |

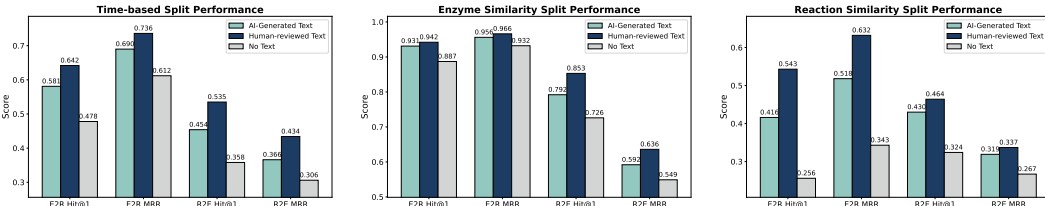

Figure 4: Performance comparison across three evaluation splits under different textual settings..

the reverse direction, yielding improvements of nearly fourfold over the best baseline. Such stability across all three splits confirms the generalization capacity and transferability of TIGER under stringent distribution shifts.

The performance gains of TIGER are more clearly illustrated in Figure 1. For brevity, Table 1 reports only Hit@1 and MRR, while a more comprehensive evaluation, including Hit@K at multiple cutoffs, Precision@K, and Mean Rank, is deferred to the Appendix, where we provide a detailed analysis across diverse evaluation dimensions.

### 5.2.2 Effect Analysis of Textual Knowledge and Dynamic Gating Network

**Effect of Textual Knowledge.** To further investigate the impact of text and its quality, we evaluated three settings: *AI-generated text* from ESM2Text, *human-reviewed text* from SwissProt, and *no text*. Since SwissProt annotations could be considered as additional resources, the results reported in the main comparison rely on AI-generated descriptions. As shown in Figure 4, incorporating text consistently improves retrieval across all splits. In the Time-based split (E2R), Hit@1 increases from 0.478 (no text) to 0.581 (AI) and 0.642 (human), while MRR rises from 0.612 → 0.690 → 0.736. Similar trends appear in R2E, with Hit@1 improving from 0.358 to 0.454 and 0.535. In the Enzyme Similarity split, both text types bring further gains, with human-reviewed text reaching 0.942 Hit@1 for E2R compared to 0.931 (AI) and 0.887 (no text). The Reaction Similarity split shows the largest relative gap: E2R Hit@1 improves from 0.256 to 0.416 and 0.543, and R2E from 0.324 to 0.430 and 0.464. These results confirm that textual information provides complementary semantics beyond sequence features, and higher-quality human-curated text delivers consistent additional benefits.

**Effect of Dynamic Gating Network.** The results in Table 2 highlight the significant contribution of the Dynamic Gating Network (DGN). Under the AI-generated text setting, DGN brings consistent gains across all metrics: for example, in the time-based split, Hit@1 improves from 0.531 to 0.581 (+0.050) and R2E Hit@1 from 0.395 to 0.454 (+0.059); in the enzyme similarity split, E2R Hit@1 rises from 0.912 to 0.931 (+0.019), and R2E MRR from 0.566 to 0.592 (+0.026). The trend is similar with SwissProt (human-reviewed) text, where DGN further strengthens performance: in the time-based split, E2R Hit@1 improves from 0.572 to 0.642 (+0.070) and R2E MRR from 0.376 to 0.434 (+0.058); in the reaction similarity split, E2R MRR increases from 0.504 to 0.632 (+0.128), repre-

Table 2: Performance comparison of with/without Dynamic Gating Network (DGN) under AI-generated and human-reviewed textual settings across three evaluation splits.

| Settings | Time-based Split | | | | Enzyme Similarity-based Split | | | | Reaction Similarity-based Split | | | |
| --- | --- | --- | --- | --- | --- | --- | --- | --- | --- | --- | --- | --- |
| | E2R | | R2E | | E2R | | R2E | | E2R | | R2E | |
| | Hit@1 | MRR | Hit@1 | MRR | Hit@1 | MRR | Hit@1 | MRR | Hit@1 | MRR | Hit@1 | MRR |
| Generated Text w/ DGN | 0.581 | 0.690 | 0.454 | 0.366 | 0.931 | 0.956 | 0.792 | 0.592 | 0.416 | 0.518 | 0.430 | 0.319 |
| Generated Text w/o DGN | 0.531 | 0.646 | 0.395 | 0.319 | 0.912 | 0.945 | 0.760 | 0.566 | 0.391 | 0.482 | 0.389 | 0.296 |
| SwissProt Text w/ DGN | 0.642 | 0.736 | 0.535 | 0.434 | 0.942 | 0.966 | 0.853 | 0.636 | 0.543 | 0.632 | 0.464 | 0.337 |
| SwissProt Text w/o DGN | 0.572 | 0.692 | 0.456 | 0.376 | 0.928 | 0.958 | 0.802 | 0.601 | 0.403 | 0.504 | 0.427 | 0.346 |

senting a substantial gain. These consistent improvements confirm that DGN effectively suppresses noisy or redundant textual signals and adaptively emphasizes informative cues, thereby enabling the model to mine richer semantic knowledge from text and achieve more robust retrieval performance.

### 5.2.3 SENSITIVITY ANALYSIS OF $\mathcal{L}_r$

We further investigate the influence of the balancing coefficient $\gamma$ on the retrieval objective $\mathcal{L}_r$ by conducting a sensitivity analysis on the time-based split. As depicted in Figure 5, the performance remains relatively stable when $\gamma$ lies within a moderate range, whereas extreme values cause pronounced degradation. In particular, assigning balanced weights to both retrieval directions ($\gamma \in [0.3, 0.7]$) leads to consistently superior results, with $\gamma = 0.7$ achieving the highest enzyme-to-reaction Hit@1 score of $0.593$. In contrast, skewed weighting substantially compromises the opposite retrieval direction: for example, $\gamma = 1.0$ maximizes the enzyme-to-reaction branch but

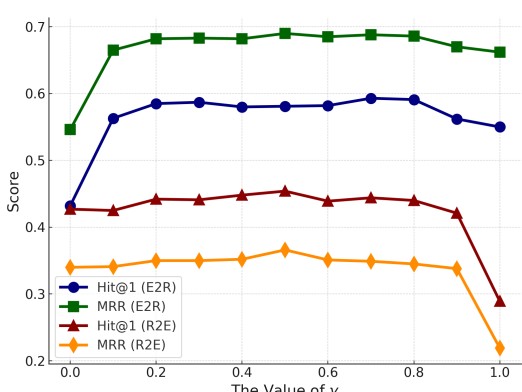

Figure 5: Sensitivity of retrieval performance with respect to the balancing parameter $\gamma$ on the time-based split.

reduces the reaction-to-enzyme Hit@1 score drastically to $0.289$. These results highlight the necessity of maintaining bidirectional balance in $\mathcal{L}_r$, and a near-symmetric weighting scheme proves preferable for robust retrieval performance. Therefore, we adopt $\gamma = 0.5$ as the default setting in all subsequent comparisons, which not only provides a fair balance between the two retrieval directions but also enhances the stability and generalization of the model across different evaluation scenarios.

## 6 CONCLUSION

In this work, we introduced TIGER, a text-informed generalized enzyme–reaction retrieval framework that addresses the fundamental challenges of directional asymmetry and distributional sensitivity in existing approaches. By leveraging knowledge-rich textual descriptions generated from protein-to-text generation models, TIGER augments sequential representations with functional semantics, while the proposed Dynamic Gating Network ensures reliable integration by suppressing noisy or spurious textual cues. In addition, the Structure-Shared Feature Projector provides a unified embedding space that enhances cross-modal alignment and supports robust bidirectional retrieval. Comprehensive experiments on the ReactZyme benchmark demonstrate that TIGER consistently surpasses strong baselines across time-based, enzyme similarity–based, and reaction similarity–based splits, achieving both improved absolute performance and greater bidirectional consistency. Beyond state-of-the-art results, TIGER highlights the potential of text-informed paradigms for advancing biochemical retrieval tasks. In future work, we plan to extend TIGER towards more fine-grained catalytic annotations, integrate curated biochemical ontologies for richer textual supervision, and explore its applicability in related domains such as metabolic pathway analysis and protein design.

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

APPENDIX

## A  USE OF LARGE LANGUAGE MODELS (LLMS)

During the preparation of this manuscript, we made limited use of large language models (LLMs) to improve linguistic presentation. LLMs were occasionally consulted to refine wording, adjust phrasing, and enhance readability. All core components of this work, including the research ideas, methodological design, experimental setup, and analysis, were entirely developed and executed by the authors. The involvement of LLMs was strictly confined to language refinement and did not affect the scientific content, technical contributions, or conclusions of the paper.

## B  ETHICS STATEMENT

This study is based exclusively on publicly available biochemical datasets, including ReactZyme and SwissProt, which contain curated information about enzymes and reactions without any personally identifiable or sensitive data. No human or animal subjects are involved. The research is conducted solely for advancing enzyme–reaction retrieval in scientific and educational contexts, and all experiments were designed to ensure transparency, reproducibility, and responsible use of resources.

## C  REPRODUCIBILITY STATEMENT

We have taken several steps to ensure the reproducibility of our work. All implementation details, including model architectures, hyperparameter settings, training schedules, and evaluation metrics, are provided in the main text and Appendix. The full implementation, along with processed datasets and detailed instructions for training and evaluation, will be released as open-source code upon publication, enabling independent researchers to reproduce our results with minimal effort.

## D  ANALYSIS OF AI-GENERATED VS. GROUND TRUTH TEXTUAL KNOWLEDGE

To systematically evaluate the quality of AI-generated textual knowledge, we conducted a comparative analysis against curated ground truth annotations. Several general observations emerge from this comparison.

**Content completeness.** AI-generated descriptions usually capture the high-level functional role of proteins, such as indicating enzymatic classes or broad biological processes. However, they often omit key biochemical details, such as specific substrates, cofactors, or reaction directions, which are consistently present in curated ground truth annotations. A quantitative analysis confirms this trend: among more than 190,000 generated entries, only 21 descriptions were entirely missing, accounting for less than 0.02%. This indicates that the AI system achieves near-universal coverage, though the descriptions may vary in specificity and reliability.

**Terminological precision.** The AI outputs tend to favor generic terminology (e.g., "transferase," "polymerase," "biosynthesis"), whereas ground truth entries employ highly precise nomenclature, explicitly naming molecules like acetyl-CoA or MurNAc-pentapeptide. This difference highlights the tendency of generative models to produce fluent but underspecified statements.

**Readability vs. mechanistic accuracy.** The AI-generated knowledge is concise and highly readable, which makes it suitable for large-scale representation learning and integration into multimodal pipelines. Ground truth descriptions, although more complex and information-dense, provide the mechanistic detail necessary for pathway reconstruction, enzymatic mechanism studies, and precise annotation tasks.

**Application value.** These findings suggest that AI-generated descriptions can be reliably employed for weak supervision and data augmentation in large-scale learning, but they need to be complemented by curated annotations in high-stakes applications requiring mechanistic fidelity.

## D.1 Case Study Analyses

We now discuss representative cases to illustrate the relative strengths and weaknesses of AI-generated knowledge.

**Case 1: O64792.** AI: *"Required for assembly of c-type cytochromes."* Ground truth: *"Part of the complex catalyzing the transfer of heme groups to c-type cytochromes."* The AI captures the functional outcome (assembly) but omits the catalytic mechanism and substrate. This represents a typical case of under-specification.

**Case 2: A9B9W1.** AI: *"Catalyzes the transfer of the phosphoribosyl group."* Ground truth: *"Catalyzes the transfer of the phosphoribosyl group to histidine, forming phosphoribosyl-histidine."* Here, the AI description is factually correct but incomplete, lacking the explicit substrate and product. This example highlights how AI tends to truncate biochemical detail.

**Case 3: Q8A8H2.** Both AI and ground truth converge on the same description: *"Peptidoglycan polymerase that catalyzes glycan chain elongation."* This case demonstrates that for canonical and frequently studied enzymes, the AI can reproduce ground truth faithfully.

**Case 4: Q9C4Z4.** AI: *"Part of the ACDS complex that catalyzes the reaction."* Ground truth: *"Part of the ACDS complex that catalyzes the reversible cleavage of acetyl-CoA into smaller units."* The AI identifies the enzymatic complex but omits the key substrate and reaction direction. This abstraction illustrates the risk of losing mechanistic specificity.

**Case 5: Q7U336.** AI: *"Cell wall formation. Catalyzes the transfer of precursors."* Ground truth: *"Cell wall formation. Catalyzes the transfer of MurNAc-pentapeptide to lipid carriers during peptidoglycan biosynthesis."* The AI provides a useful but vague summary, whereas the ground truth supplies precise biochemical participants. This difference is crucial for pathway-level analysis.

**Case 6: Q5L5X8.** AI: *"Involved in amino acid biosynthesis."* Ground truth: *"Catalyzes the condensation of aspartate-semialdehyde and homoserine to form threonine."* The AI description captures the correct pathway but lacks mechanistic clarity, showing how generated text favors abstraction over specificity.

**Case 7: P76218.** AI: *"Functions in DNA repair."* Ground truth: *"DNA glycosylase that excises uracil from DNA to initiate base-excision repair."* The AI is directionally correct but insufficient for mechanistic studies. Ground truth adds the enzymatic role and specific target, which are indispensable for accurate interpretation.

**Case 8: B1XQJ8.** AI: *"Plays a role in metabolic adaptation."* Ground truth: *"Catalyzes the reversible interconversion of malate and oxaloacetate as part of the TCA cycle."* Here, the AI description is vague to the point of being biologically uninformative, while the ground truth situates the enzyme within a well-defined metabolic context. This represents a case where AI abstraction risks undermining downstream interpretability.

In summary, these case studies reveal a recurring pattern: AI-generated knowledge excels at producing concise and standardized statements that capture overarching functional roles, and its coverage is nearly complete (with fewer than 0.02% missing descriptions across the dataset). Such scalability makes AI-generated text highly valuable for large-scale representation learning and weakly supervised annotation. Nevertheless, curated ground truth provides indispensable mechanistic precision by specifying substrates, products, cofactors, and reaction dynamics. Without these details, downstream biological analyses risk oversimplification or misinterpretation. We therefore advocate a complementary use of the two sources: AI-generated text for breadth, scalability, and uniformity, and ground truth for biochemical rigor and reproducibility. Future research should explore hybrid frameworks that leverage AI outputs for hypothesis generation while grounding high-stakes applications in curated annotations, thereby achieving both scalability and accuracy in knowledge-driven modeling.

# E    EVALUATION METRICS

We adopted several widely used retrieval metrics to comprehensively evaluate model performance:

**Hit@K.**   Hit@K measures whether the ground-truth item appears within the top-$K$ retrieved results. Formally, for a query $q$, let $\mathrm{rank}(q)$ denote the rank position of its ground-truth item. Then

$$\mathrm{Hit@K} = \frac{1}{N} \sum_{i=1}^{N} \mathbb{I}\big[\mathrm{rank}(q_i) \leq K\big],$$

where $N$ is the total number of queries. Hit@K is also equivalent to Top-$K$ accuracy in classification settings.

**Precision@K.**   Precision@K evaluates how many of the retrieved top-$K$ items are correct. This metric is particularly useful when a query may correspond to multiple valid ground-truth items. It is defined as

$$\mathrm{Precision@K} = \frac{1}{N} \sum_{i=1}^{N} \frac{|\mathrm{Retrieved}(q_i, K) \cap \mathrm{GT}(q_i)|}{K},$$

where $\mathrm{Retrieved}(q_i, K)$ is the set of top-$K$ results for query $q_i$, and $\mathrm{GT}(q_i)$ is its ground-truth set.

**Mean Reciprocal Rank (MRR).**   MRR evaluates ranking quality by rewarding higher scores when the correct item appears earlier in the ranked list:

$$\mathrm{MRR} = \frac{1}{N} \sum_{i=1}^{N} \frac{1}{\mathrm{rank}(q_i)}.$$

**Mean Rank (MR).**   Mean Rank directly computes the average rank position of the ground-truth items:

$$\mathrm{MR} = \frac{1}{N} \sum_{i=1}^{N} \mathrm{rank}(q_i).$$

Lower values indicate better performance, as the correct items tend to appear earlier in the ranking.

# F    BASELINES UNDER THE REACTZYME PROTOCOL

We evaluate our method against representative baselines implemented under the **ReactZyme** protocol for bidirectional enzyme-reaction retrieval. Unless otherwise stated, all methods adopt a dual-encoder architecture with a temperature-scaled cosine similarity and a symmetric InfoNCE-style contrastive objective for both directions (E→R and R→E). Superscripts in Table 1 indicate the specific encoder pairing used by each baseline: [1] UniMol-3D (reaction) + ESM (enzyme), [2] MAT-2D (reaction) + ESM (enzyme), [3] UniMol-3D (reaction) + SaProt (enzyme).

**ReactZyme (Base).**   The foundational benchmark that establishes the training/evaluation protocol for enzyme-reaction retrieval. On the enzyme side, a protein language model (ESM or SaProt per the superscript) encodes amino-acid sequences; on the reaction side, a learned chemical encoder (UniMol-3D or MAT-2D per the superscript) produces reaction embeddings. Both embeddings are projected to a shared space via lightweight MLP heads and trained with a bidirectional contrastive loss using in-batch negatives.

**Fingerprint.**   A non-neural reaction representation baseline where reactions are encoded by standard chemical fingerprints (RDKit). The enzyme encoder follows the ReactZyme setup (ESM or SaProt depending on the variant), and a small MLP is used to map both sides into the shared space. This baseline is computationally efficient but typically less expressive for complex reaction semantics.

**GNN**[1,3]. A graph-neural reaction encoder variant in which molecular graphs (and available 3D cues) are processed by a message-passing backbone with a graph-level readout. In our runs, the reaction side corresponds to UniMol-3D ([1,3]), paired with ESM ([1]) or SaProt ([3]) on the enzyme side as indicated in the table. Projection heads and the contrastive training recipe remain identical to ReactZyme.

**Bi–RNN**[1,2]. As another sequential decoding baseline, we consider a bidirectional recurrent neural network (Bi-RNN) as the decoder. Unlike the simple feed-forward MLP, the Bi-RNN is designed to capture temporal dependencies by processing the encoded representations in both forward and backward directions, thereby modeling contextual interactions across the sequence. In our implementation, the Bi-RNN decoder takes the projected embeddings as input and produces hidden states that are aggregated to form the final retrieval scores. This architecture allows the model to exploit sequential ordering and long-range dependencies, providing a stronger sequential inductive bias compared to the MLP decoder. According to the ReactZyme paper (Hua et al., 2024), the Bi-RNN decoder demonstrated the best retrieval performance among the tested alternatives; therefore, in our main experiments we adopt this variant for comparison.

**CLIPZyme**[1,2]. CLIPZyme Mikhael et al. (2024) is a CLIP-style dual-encoder framework designed for enzyme-reaction retrieval. On the enzyme side, it adopts a protein language model encoder (e.g., ESM), while on the reaction side, it introduces a novel representation by constructing a *pseudo-transition state graph* that connects substrates and products. This graph is intended to approximate the intermediate transition state of biochemical reactions, thereby enriching the reaction representation beyond standard molecular encodings. Both enzyme and reaction embeddings are projected into a shared latent space, and training is conducted using a contrastive objective similar to the CLIP paradigm. In Table 1, superscripts [1] and [2] indicate the use of UniMol-3D or MAT-2D as the reaction encoder in place of the pseudo-transition graph for ablation-style comparisons under the ReactZyme protocol.

### F.1 EXTRA EXPERIMENTS AND ANALYSIS

To further validate the robustness and generalizability of our framework, we provide an extended set of experiments across three evaluation splits: the *time-based split*, the *enzyme similarity-based split*, and the *reaction similarity-based split*. These additional experiments not only complement the main results presented in the previous section but also offer deeper insights into how TIGER performs under different levels of distributional shifts. For each split, we analyze both retrieval directions (enzyme-to-reaction and reaction-to-enzyme) with multiple evaluation metrics, including Hit@k, Precision@k, Mean Reciprocal Rank (MRR), and Mean Rank. The following subsections summarize the results and provide detailed analyses.

### F.2 ANALYSIS OF RETRIEVAL PERFORMANCE ON TIME-BASED SPLITS

**Enzyme-to-Reaction Retrieval (Hit@k and MRR).** From Table 3, we observe that TIGER markedly outperforms all baseline methods across all cutoff thresholds and in terms of MRR. Specifically, TIGER achieves a Hit@1 of **0.5810**, representing a relative improvement of nearly 48% over the strongest baseline (Bi-RNN[2], 0.3911). Similar improvements persist at higher cutoff thresholds: for example, at Hit@20, TIGER reaches **0.9164**, which significantly exceeds the next-best baseline (Bi-RNN[2], 0.8559). The consistent margins across H@k levels highlight that TIGER is not only more accurate at top-1 retrieval but also ensures stable ranking quality deeper into the candidate list. Moreover, the MRR of **0.6902** represents a substantial leap over the baseline range (0.2788-0.5303), further validating TIGER's effectiveness in optimizing rank-sensitive metrics.

**Enzyme-to-Reaction Retrieval (Precision@k and Mean Rank).** As shown in Table 4, TIGER sustains its advantage when evaluated with precision-oriented metrics. At P@1, TIGER again attains **0.5810**, clearly outperforming Bi-RNN[2] (0.3911). While precision values naturally decay with larger k, TIGER consistently dominates baselines across all cutoffs. More importantly, TIGER achieves a **mean rank of 13.33**, which is dramatically lower (better) than those of existing methods, where even the strongest baselines remain above 30-40. This indicates that correct reactions for

Table 3: Enzyme to Reaction Retrieval Performance (H@k and MRR) on Time-based Split.

| Method | H@1 | H@2 | H@3 | H@4 | H@5 | H@10 | H@20 | MRR |
|---|---|---|---|---|---|---|---|---|
| ReactZyme[1] | 0.2905 | 0.4007 | 0.4563 | 0.4984 | 0.5365 | 0.6586 | 0.7639 | 0.4104 |
| ReactZyme[2] | 0.3246 | 0.4526 | 0.5255 | 0.5700 | 0.6044 | 0.7079 | 0.7972 | 0.4549 |
| ReactZyme[3] | 0.0916 | 0.1328 | 0.1650 | 0.1908 | 0.2134 | 0.2923 | 0.3882 | 0.2788 |
| Fingerprint | 0.2357 | 0.3470 | 0.3968 | 0.4215 | 0.4684 | 0.5439 | 0.7040 | 0.2984 |
| GNN[1] | 0.3588 | 0.5158 | 0.5919 | 0.6044 | 0.6545 | 0.7815 | 0.8126 | 0.4952 |
| GNN[3] | 0.2508 | 0.3528 | 0.3995 | 0.4016 | 0.4075 | 0.5448 | 0.6421 | 0.3453 |
| Bi-RNN[1] | 0.3543 | 0.5112 | 0.5820 | 0.6250 | 0.6563 | 0.7480 | 0.8259 | 0.4946 |
| Bi-RNN[2] | 0.3911 | 0.5542 | 0.6170 | 0.6555 | 0.6875 | 0.7847 | 0.8559 | 0.5303 |
| CLIPZyme[1] | 0.2631 | 0.3670 | 0.4189 | 0.4447 | 0.4534 | 0.6444 | 0.7516 | 0.3940 |
| CLIPZyme[2] | 0.3041 | 0.4346 | 0.4991 | 0.5610 | 0.5993 | 0.6943 | 0.7840 | 0.4355 |
| TIGER (Ours) | **0.5810** | **0.7187** | **0.7678** | **0.7972** | **0.8190** | **0.8740** | **0.9164** | **0.6902** |

enzymes are not only placed earlier but are much more concentrated toward the top of the retrieval list under TIGER's ranking.

Table 4: Enzyme to Reaction Retrieval Performance (P@k and Mean Rank) on Time-based Split.

| Method | P@1 | P@2 | P@3 | P@4 | P@5 | P@10 | P@20 | Mean Rank |
|---|---|---|---|---|---|---|---|---|
| ReactZyme[1] | 0.2905 | 0.2004 | 0.1522 | 0.1247 | 0.1074 | 0.0659 | 0.0382 | 46.0553 |
| ReactZyme[2] | 0.3246 | 0.2263 | 0.1752 | 0.1425 | 0.1209 | 0.0708 | 0.0399 | 40.4756 |
| ReactZyme[3] | 0.0916 | 0.0664 | 0.0550 | 0.0477 | 0.0426 | 0.0292 | 0.0194 | 168.8244 |
| Fingerprint | 0.2357 | 0.1736 | 0.1323 | 0.1054 | 0.0937 | 0.0544 | 0.0352 | 89.5675 |
| GNN[1] | 0.3588 | 0.2579 | 0.1973 | 0.1511 | 0.1309 | 0.0781 | 0.0406 | 32.7443 |
| GNN[3] | 0.2508 | 0.1764 | 0.1331 | 0.1004 | 0.0815 | 0.0546 | 0.0321 | 59.8345 |
| Bi-RNN[1] | 0.3543 | 0.2556 | 0.1940 | 0.1563 | 0.1313 | 0.0748 | 0.0413 | 34.6103 |
| Bi-RNN[2] | 0.3911 | 0.2771 | 0.2057 | 0.1639 | 0.1375 | 0.0785 | 0.0428 | 35.2791 |
| CLIPZyme[1] | 0.2631 | 0.1835 | 0.1401 | 0.1112 | 0.0907 | 0.0645 | 0.0376 | 45.3637 |
| CLIPZyme[2] | 0.3041 | 0.2173 | 0.1658 | 0.1399 | 0.1201 | 0.0695 | 0.0392 | 42.3645 |
| TIGER (Ours) | **0.5810** | **0.3593** | **0.2559** | **0.1993** | **0.1638** | **0.0874** | **0.0458** | **13.3309** |

**Reaction-to-Enzyme Retrieval (Hit@k and MRR).**   Table 5 demonstrates similar trends in the reverse retrieval direction. TIGER surpasses all baselines substantially, with a Hit@1 of **0.4536**, compared to 0.2650 for Bi-RNN[2], the closest competitor. The improvements remain consistent as k increases: TIGER achieves **0.6708** at Hit@5 and **0.8477** at Hit@20, surpassing the best baselines by wide margins. Although absolute values are slightly lower than in the enzyme-to-reaction setting (reflecting the greater difficulty of this direction), TIGER still provides strong improvements in MRR (**0.3658** versus 0.2267), highlighting its robust generalization across both retrieval tasks.

**Reaction-to-Enzyme Retrieval (Precision@k and Mean Rank).**   Finally, Table 6 shows TIGER's precision and ranking performance in the reaction-to-enzyme direction. TIGER achieves a P@1 of **0.4536**, outperforming Bi-RNN[2] (0.2650) by more than 70%. The performance gap persists across increasing cutoffs, indicating that TIGER can consistently identify relevant enzymes even when more candidates are considered. Crucially, TIGER's **mean rank of 45.13** represents a major reduction compared to the 138-700 range of baselines, showing that TIGER drastically shortens the search depth required to find correct enzymes.

### F.3  ANALYSIS OF RETRIEVAL PERFORMANCE ON ENZYME SIMILARITY-BASED SPLITS

**Enzyme-to-Reaction Retrieval (Hit@k and MRR).**   From Table 7, TIGER achieves substantial improvements across all cutoff thresholds. For instance, TIGER attains a Hit@1 of **0.9308**, significantly higher than the strongest baseline Bi-RNN[2] (0.8151), marking a relative gain of over 14%.

Table 5: Reaction to Enzyme Retrieval Performance (H@k and MRR) on Time-based Split.

| Method | H@1 | H@2 | H@3 | H@4 | H@5 | H@10 | H@20 | MRR |
|---|---|---|---|---|---|---|---|---|
| ReactZyme[1] | 0.1678 | 0.2240 | 0.2631 | 0.2938 | 0.3155 | 0.3960 | 0.5011 | 0.1400 |
| ReactZyme[2] | 0.2175 | 0.2733 | 0.3144 | 0.3493 | 0.3815 | 0.4924 | 0.6033 | 0.1789 |
| ReactZyme[3] | 0.0558 | 0.0721 | 0.0815 | 0.0883 | 0.0979 | 0.1359 | 0.1918 | 0.0538 |
| Fingerprint | 0.1435 | 0.2017 | 0.2345 | 0.2656 | 0.2980 | 0.3547 | 0.4582 | 0.1166 |
| GNN[1] | 0.2045 | 0.2835 | 0.3398 | 0.3722 | 0.3792 | 0.4475 | 0.5168 | 0.1628 |
| GNN[3] | 0.1331 | 0.1750 | 0.1886 | 0.1979 | 0.2044 | 0.3365 | 0.4119 | 0.1122 |
| Bi-RNN[1] | 0.2540 | 0.3261 | 0.3747 | 0.4024 | 0.4324 | 0.5330 | 0.6481 | 0.2113 |
| Bi-RNN[2] | 0.2650 | 0.3470 | 0.3994 | 0.4355 | 0.4704 | 0.5854 | 0.6940 | 0.2267 |
| CLIPZyme[1] | 0.1331 | 0.2034 | 0.2451 | 0.2822 | 0.2993 | 0.3554 | 0.4567 | 0.1313 |
| CLIPZyme[2] | 0.1757 | 0.2445 | 0.3062 | 0.3075 | 0.3447 | 0.4555 | 0.5343 | 0.1678 |
| TIGER (Ours) | **0.4536** | **0.5474** | **0.6025** | **0.6427** | **0.6708** | **0.7676** | **0.8477** | **0.3658** |

Table 6: Reaction to Enzyme Retrieval Performance (P@k and Mean Rank) on Time-based Split.

| Method | P@1 | P@2 | P@3 | P@4 | P@5 | P@10 | P@20 | Mean Rank |
|---|---|---|---|---|---|---|---|---|
| ReactZyme[1] | 0.1678 | 0.1543 | 0.1443 | 0.1349 | 0.1267 | 0.1002 | 0.0748 | 177.4881 |
| ReactZyme[2] | 0.2175 | 0.2001 | 0.1817 | 0.1688 | 0.1570 | 0.1206 | 0.0871 | 165.3066 |
| ReactZyme[3] | 0.0558 | 0.0497 | 0.0448 | 0.0407 | 0.0393 | 0.0344 | 0.0278 | 700.9714 |
| Fingerprint | 0.1435 | 0.1212 | 0.1147 | 0.1039 | 0.1031 | 0.0912 | 0.0734 | 200.4936 |
| GNN[1] | 0.2045 | 0.1955 | 0.1867 | 0.1715 | 0.1523 | 0.1133 | 0.0749 | 167.5862 |
| GNN[3] | 0.1331 | 0.1207 | 0.1036 | 0.0912 | 0.0821 | 0.0852 | 0.0597 | 322.5755 |
| Bi-RNN[1] | 0.2540 | 0.2270 | 0.2065 | 0.1875 | 0.1731 | 0.1323 | 0.0949 | 138.5832 |
| Bi-RNN[2] | 0.2650 | 0.2399 | 0.2202 | 0.2030 | 0.1892 | 0.1451 | 0.1028 | 149.2686 |
| CLIPZyme[1] | 0.1331 | 0.1417 | 0.1250 | 0.1149 | 0.1033 | 0.0949 | 0.0740 | 186.4576 |
| CLIPZyme[2] | 0.1757 | 0.1630 | 0.1532 | 0.1443 | 0.1312 | 0.1101 | 0.0756 | 173.3521 |
| TIGER (Ours) | **0.4536** | **0.3936** | **0.3486** | **0.3126** | **0.2853** | **0.2016** | **0.1328** | **45.1359** |

The margins remain consistent at higher cutoffs, with TIGER reaching **0.9962** at Hit@20, compared to 0.9913 for Bi-RNN[2]. Furthermore, TIGER's MRR of **0.9561** exceeds all baselines by a large margin, demonstrating its superior ability to prioritize the correct reaction in top ranks. These results highlight that TIGER is highly effective even when training and test enzymes are evolutionarily distant.

Table 7: Enzyme to Reaction Retrieval Performance (H@k and MRR) on Enzyme Similarity-based Split.

| Method | H@1 | H@2 | H@3 | H@4 | H@5 | H@10 | H@20 | MRR |
|---|---|---|---|---|---|---|---|---|
| ReactZyme[1] | 0.7267 | 0.8366 | 0.8758 | 0.9002 | 0.9062 | 0.9487 | 0.9632 | 0.8112 |
| ReactZyme[2] | 0.5987 | 0.7737 | 0.8311 | 0.8650 | 0.8759 | 0.9328 | 0.9572 | 0.7280 |
| ReactZyme[3] | 0.5998 | 0.7592 | 0.8164 | 0.8522 | 0.8665 | 0.9229 | 0.9454 | 0.7226 |
| Fingerprint | 0.5790 | 0.6507 | 0.7240 | 0.8230 | 0.7743 | 0.9169 | 0.8700 | 0.6393 |
| GNN[1] | 0.7111 | 0.8273 | 0.8668 | 0.8798 | 0.9017 | 0.9547 | 0.9592 | 0.8023 |
| GNN[3] | 0.6328 | 0.8002 | 0.8077 | 0.8790 | 0.8853 | 0.9348 | 0.9513 | 0.7457 |
| Bi-RNN[1] | 0.8114 | 0.9014 | 0.9287 | 0.9413 | 0.9503 | 0.9731 | 0.9851 | 0.8747 |
| Bi-RNN[2] | 0.8151 | 0.9260 | 0.9532 | 0.9629 | 0.9713 | 0.9850 | 0.9913 | 0.8861 |
| CLIPZyme[1] | 0.7547 | 0.8706 | 0.9105 | 0.9642 | 0.9478 | 0.9679 | 0.9780 | 0.8546 |
| CLIPZyme[2] | 0.5489 | 0.6851 | 0.7351 | 0.7970 | 0.7768 | 0.9290 | 0.9460 | 0.6971 |
| TIGER (Ours) | **0.9308** | **0.9707** | **0.9783** | **0.9816** | **0.9850** | **0.9916** | **0.9962** | **0.9561** |

**Enzyme-to-Reaction Retrieval (Precision@k and Mean Rank).** Table 8 further confirms TIGER's advantage from a precision and ranking perspective. At P@1, TIGER reaches **0.9308**, which is markedly higher than Bi-RNN[2] (0.8151). Although precision naturally decreases as $k$ increases, TIGER consistently maintains the highest values across all cutoffs. Importantly, the mean rank drops to only **1.58**, far better than the best baseline (2.71 for Bi-RNN[2]). This indicates that TIGER almost always positions the correct reaction within the very first few retrieved candidates, yielding highly efficient retrieval.

Table 8: Enzyme to Reaction Retrieval Performance (P@k and Mean Rank) on Enzyme Similarity-based Split.

| Method | P@1 | P@2 | P@3 | P@4 | P@5 | P@10 | P@20 | Mean Rank |
|---|---|---|---|---|---|---|---|---|
| ReactZyme[1] | 0.7267 | 0.4177 | 0.2926 | 0.2248 | 0.1835 | 0.0955 | 0.0488 | 4.5799 |
| ReactZyme[2] | 0.5987 | 0.3864 | 0.2777 | 0.2160 | 0.1774 | 0.0939 | 0.0485 | 5.3021 |
| ReactZyme[3] | 0.5998 | 0.3792 | 0.2728 | 0.2128 | 0.1755 | 0.0929 | 0.0479 | 7.4701 |
| Fingerprint | 0.5790 | 0.3255 | 0.2414 | 0.2058 | 0.1549 | 0.0917 | 0.0435 | 12.4571 |
| GNN[1] | 0.7111 | 0.4131 | 0.2896 | 0.2197 | 0.1826 | 0.0961 | 0.0486 | 4.8395 |
| GNN[3] | 0.6328 | 0.3996 | 0.2699 | 0.2195 | 0.1793 | 0.0941 | 0.0482 | 6.9597 |
| Bi-RNN[1] | 0.8114 | 0.4507 | 0.3096 | 0.2354 | 0.1901 | 0.0973 | 0.0493 | 3.5925 |
| Bi-RNN[2] | 0.8151 | 0.4632 | 0.3179 | 0.2408 | 0.1943 | 0.0986 | 0.0496 | 2.7051 |
| CLIPZyme[1] | 0.7547 | 0.4355 | 0.3036 | 0.2411 | 0.1896 | 0.0968 | 0.0489 | 3.9820 |
| CLIPZyme[2] | 0.5489 | 0.3427 | 0.2451 | 0.1993 | 0.1554 | 0.0929 | 0.0473 | 8.3524 |
| TIGER (Ours) | **0.9308** | **0.4853** | **0.3261** | **0.2454** | **0.1970** | **0.0991** | **0.0498** | **1.5807** |

**Reaction-to-Enzyme Retrieval (Hit@k and MRR).** As shown in Table 9, TIGER also excels in the reverse retrieval direction. At Hit@1, TIGER obtains **0.7921**, substantially surpassing Bi-RNN[2] (0.5887), with consistent improvements at higher cutoffs (e.g., Hit@20: 0.9809 vs. 0.9669). The MRR of **0.5921** further highlights TIGER's ability to concentrate correct enzyme matches near the top of the ranked list, even when test reactions differ substantially from training examples.

Table 9: Reaction to Enzyme Retrieval Performance (H@k and MRR) on Enzyme Similarity-based Split.

| Method | H@1 | H@2 | H@3 | H@4 | H@5 | H@10 | H@20 | MRR |
|---|---|---|---|---|---|---|---|---|
| ReactZyme[1] | 0.4088 | 0.5246 | 0.5987 | 0.6480 | 0.6892 | 0.7953 | 0.8666 | 0.2930 |
| ReactZyme[2] | 0.3624 | 0.4545 | 0.5190 | 0.5697 | 0.6091 | 0.7225 | 0.7986 | 0.2586 |
| ReactZyme[3] | 0.3477 | 0.4427 | 0.5082 | 0.5522 | 0.5458 | 0.6980 | 0.7762 | 0.2563 |
| Fingerprint | 0.2545 | 0.3047 | 0.3569 | 0.4170 | 0.4686 | 0.5470 | 0.6987 | 0.2035 |
| GNN[1] | 0.3928 | 0.4910 | 0.5515 | 0.6113 | 0.6612 | 0.7628 | 0.8324 | 0.2837 |
| GNN[3] | 0.3655 | 0.4706 | 0.5187 | 0.5682 | 0.6161 | 0.7376 | 0.7552 | 0.2633 |
| Bi-RNN[1] | 0.5086 | 0.6217 | 0.6904 | 0.7470 | 0.7832 | 0.8697 | 0.9243 | 0.3869 |
| Bi-RNN[2] | 0.5887 | 0.7120 | 0.7756 | 0.8252 | 0.8551 | 0.9193 | 0.9669 | 0.4562 |
| CLIPZyme[1] | 0.3570 | 0.4835 | 0.5647 | 0.6146 | 0.6371 | 0.7552 | 0.8431 | 0.2828 |
| CLIPZyme[2] | 0.3337 | 0.4371 | 0.4835 | 0.5352 | 0.6077 | 0.6514 | 0.7687 | 0.2038 |
| TIGER (Ours) | **0.7921** | **0.8766** | **0.9116** | **0.9281** | **0.9408** | **0.9688** | **0.9809** | **0.5921** |

**Reaction-to-Enzyme Retrieval (Precision@k and Mean Rank).** Table 10 shows that TIGER maintains strong performance from a precision-oriented perspective. At P@1, TIGER achieves **0.7921**, compared to 0.5887 for Bi-RNN[2], representing an improvement of nearly 35%. The relative margins remain across P@k levels, confirming TIGER's robustness under this challenging split. Moreover, TIGER yields a mean rank of only **6.81**, dramatically lower than all baselines (the best baseline being 9.79 from Bi-RNN[2]). This demonstrates that TIGER requires far fewer ranking steps to identify the correct enzyme, making it especially advantageous for practical applications.

Table 10: Reaction to Enzyme Retrieval Performance (P@k and Mean Rank) on Enzyme Similarity-based Split.

| Method | P@1 | P@2 | P@3 | P@4 | P@5 | P@10 | P@20 | Mean Rank |
|---|---|---|---|---|---|---|---|---|
| ReactZyme[1] | 0.4088 | 0.3951 | 0.3725 | 0.3516 | 0.3350 | 0.2690 | 0.1975 | 24.2505 |
| ReactZyme[2] | 0.3624 | 0.3423 | 0.3229 | 0.3091 | 0.2961 | 0.2444 | 0.1820 | 22.5053 |
| ReactZyme[3] | 0.3477 | 0.3334 | 0.3162 | 0.2996 | 0.2653 | 0.2361 | 0.1769 | 34.9487 |
| Fingerprint | 0.2545 | 0.2436 | 0.2257 | 0.2038 | 0.2012 | 0.1847 | 0.1796 | 45.6897 |
| GNN[1] | 0.3928 | 0.3698 | 0.3431 | 0.3317 | 0.3214 | 0.2580 | 0.1897 | 23.8241 |
| GNN[3] | 0.3655 | 0.3544 | 0.3227 | 0.3083 | 0.2995 | 0.2495 | 0.1721 | 22.8901 |
| Bi-RNN[1] | 0.5086 | 0.4727 | 0.4376 | 0.4094 | 0.3851 | 0.3001 | 0.2117 | 14.7945 |
| Bi-RNN[2] | 0.5887 | 0.5318 | 0.4804 | 0.4447 | 0.4135 | 0.3110 | 0.2177 | 9.7913 |
| CLIPZyme[1] | 0.3570 | 0.3478 | 0.3212 | 0.3196 | 0.2885 | 0.2577 | 0.1834 | 25.5786 |
| CLIPZyme[2] | 0.3337 | 0.3245 | 0.3094 | 0.2971 | 0.2844 | 0.2235 | 0.1811 | 30.4196 |
| TIGER (Ours) | **0.7921** | **0.6586** | **0.5689** | **0.5071** | **0.4606** | **0.3301** | **0.2238** | **6.8100** |

### F.4 ANALYSIS OF RETRIEVAL PERFORMANCE ON ENZYME SIMILARITY-BASED SPLITS

**Enzyme-to-Reaction Retrieval (Hit@k and MRR).** From Table 11, TIGER achieves dramatic improvements over all baselines. At Hit@1, TIGER attains **0.4155**, which is nearly four times higher than the strongest baseline (CLIPZyme[1], 0.1305). The improvements remain consistent across higher cutoffs, with TIGER reaching **0.7540** at Hit@20, far exceeding the best baseline (0.6220). In terms of MRR, TIGER records **0.5180**, a substantial leap compared to baselines that remain below 0.24. These results demonstrate that TIGER can effectively prioritize correct reactions even when reaction similarity cues are absent, a scenario where existing methods struggle.

Table 11: Enzyme to Reaction Retrieval Performance (H@k and MRR) on Reaction Similarity-based Split.

| Method | H@1 | H@2 | H@3 | H@4 | H@5 | H@10 | H@20 | MRR |
|---|---|---|---|---|---|---|---|---|
| ReactZyme[1] | 0.0912 | 0.1495 | 0.2321 | 0.2177 | 0.2580 | 0.4213 | 0.4571 | 0.1856 |
| ReactZyme[2] | 0.0914 | 0.1604 | 0.2471 | 0.2694 | 0.2968 | 0.4373 | 0.5908 | 0.2005 |
| ReactZyme[3] | 0.1085 | 0.1638 | 0.2112 | 0.2257 | 0.2699 | 0.4034 | 0.5429 | 0.1988 |
| Fingerprint | 0.0935 | 0.1607 | 0.2270 | 0.2771 | 0.3004 | 0.4400 | 0.6000 | 0.1935 |
| GNN[1] | 0.1104 | 0.1691 | 0.2368 | 0.2742 | 0.3023 | 0.4573 | 0.5669 | 0.2011 |
| GNN[3] | 0.0962 | 0.1592 | 0.2265 | 0.2285 | 0.2545 | 0.4024 | 0.5289 | 0.1972 |
| Bi-RNN[1] | 0.1085 | 0.1543 | 0.1836 | 0.2177 | 0.2603 | 0.4077 | 0.5594 | 0.1969 |
| Bi-RNN[2] | 0.1181 | 0.2179 | 0.2787 | 0.3274 | 0.3664 | 0.4897 | 0.6068 | 0.2399 |
| CLIPZyme[1] | 0.1305 | 0.2392 | 0.3093 | 0.3604 | 0.3420 | 0.5320 | 0.6220 | 0.1937 |
| CLIPZyme[2] | 0.1235 | 0.2281 | 0.2912 | 0.3415 | 0.3064 | 0.5719 | 0.6000 | 0.2201 |
| TIGER (Ours) | **0.4155** | **0.5234** | **0.5812** | **0.6117** | **0.6416** | **0.6827** | **0.7540** | **0.5180** |

**Enzyme-to-Reaction Retrieval (Precision@k and Mean Rank).** As shown in Table 12, TIGER achieves the highest precision across all cutoff levels. At P@1, TIGER reaches **0.4155**, far surpassing the best baseline (CLIPZyme[1], 0.1305). Although precision decreases as $k$ increases, TIGER consistently maintains a considerable margin over all alternatives. Most notably, the mean rank of TIGER is only **23.25**, compared to the next best value of 35.65 (CLIPZyme[2]) and much higher values exceeding 90 for weaker baselines. This indicates that TIGER retrieves correct reactions much earlier in the ranking process, a critical advantage for practical applications.

**Reaction-to-Enzyme Retrieval (Hit@k and MRR).** Table 13 shows that TIGER continues to outperform baselines in the reverse retrieval direction. TIGER achieves a Hit@1 of **0.4305**, significantly higher than Bi-RNN[2] (0.1710) or CLIPZyme[2] (0.1457). At Hit@20, TIGER maintains strong performance with **0.7616**, compared to 0.5855 for Bi-RNN[2]. The MRR of **0.3185** further

Table 12: Enzyme to Reaction Retrieval Performance (P@k and Mean Rank) on Reaction Similarity-based Split.

| Method | P@1 | P@2 | P@3 | P@4 | P@5 | P@10 | P@20 | Mean Rank |
|---|---|---|---|---|---|---|---|---|
| ReactZyme[1] | 0.0912 | 0.0752 | 0.0699 | 0.0547 | 0.0518 | 0.0422 | 0.0229 | 92.2778 |
| ReactZyme[2] | 0.0914 | 0.0807 | 0.0744 | 0.0677 | 0.0596 | 0.0438 | 0.0296 | 39.9146 |
| ReactZyme[3] | 0.1085 | 0.0824 | 0.0636 | 0.0567 | 0.0542 | 0.0404 | 0.0272 | 42.3597 |
| Fingerprint | 0.0935 | 0.0804 | 0.0757 | 0.0693 | 0.0601 | 0.0440 | 0.0300 | 45.3825 |
| GNN[1] | 0.1104 | 0.0851 | 0.0713 | 0.0689 | 0.0607 | 0.0458 | 0.0284 | 38.9685 |
| GNN[3] | 0.0962 | 0.0801 | 0.0682 | 0.0574 | 0.0511 | 0.0403 | 0.0265 | 50.9663 |
| Bi-RNN[1] | 0.1181 | 0.1090 | 0.0929 | 0.0819 | 0.0733 | 0.0490 | 0.0303 | 41.3776 |
| Bi-RNN[2] | 0.1085 | 0.0771 | 0.0612 | 0.0544 | 0.0521 | 0.0408 | 0.0280 | 41.3069 |
| CLIPZyme[1] | 0.1305 | 0.1196 | 0.1031 | 0.0901 | 0.0684 | 0.0532 | 0.0311 | 48.4672 |
| CLIPZyme[2] | 0.1235 | 0.1146 | 0.0971 | 0.0854 | 0.0613 | 0.0572 | 0.0300 | 35.6457 |
| TIGER (Ours) | **0.4155** | **0.2617** | **0.1937** | **0.1529** | **0.1283** | **0.0682** | **0.0377** | **23.2479** |

demonstrates TIGER's capacity to bring relevant enzymes much closer to the top of the ranked list, substantially improving over all baselines that remain below 0.17.

Table 13: Reaction to Enzyme Retrieval Performance (H@k and MRR) on Reaction Similarity-based Split.

| Method | H@1 | H@2 | H@3 | H@4 | H@5 | H@10 | H@20 | MRR |
|---|---|---|---|---|---|---|---|---|
| ReactZyme[1] | 0.0924 | 0.1063 | 0.1208 | 0.1277 | 0.1332 | 0.1790 | 0.2172 | 0.0943 |
| ReactZyme[2] | 0.1347 | 0.1622 | 0.1812 | 0.1835 | 0.2000 | 0.2326 | 0.2753 | 0.1341 |
| ReactZyme[3] | 0.0933 | 0.1274 | 0.1478 | 0.1617 | 0.1703 | 0.2130 | 0.2613 | 0.0962 |
| Fingerprint | 0.1143 | 0.1346 | 0.1514 | 0.1650 | 0.1774 | 0.1829 | 0.2325 | 0.1042 |
| GNN[1] | 0.1244 | 0.1573 | 0.1735 | 0.1867 | 0.2058 | 0.2440 | 0.2848 | 0.1129 |
| GNN[3] | 0.0917 | 0.1100 | 0.1219 | 0.1312 | 0.1418 | 0.1847 | 0.2234 | 0.1051 |
| Bi-RNN[1] | 0.1244 | 0.1813 | 0.2150 | 0.2383 | 0.2565 | 0.3990 | 0.4948 | 0.1206 |
| Bi-RNN[2] | 0.1710 | 0.2254 | 0.2694 | 0.3187 | 0.3549 | 0.4741 | 0.5855 | 0.1696 |
| CLIPZyme[1] | 0.1298 | 0.1573 | 0.1799 | 0.1842 | 0.1993 | 0.2215 | 0.2544 | 0.1245 |
| CLIPZyme[2] | 0.1457 | 0.1741 | 0.1905 | 0.1944 | 0.2173 | 0.2456 | 0.2893 | 0.1521 |
| TIGER (Ours) | **0.4305** | **0.5181** | **0.5595** | **0.5906** | **0.6113** | **0.6994** | **0.7616** | **0.3185** |

**Reaction-to-Enzyme Retrieval (Precision@k and Mean Rank).** Table 14 further highlights TIGER's robustness. At P@1, TIGER reaches **0.4305**, outperforming the best baseline (Bi-RNN[2], 0.1710) by a wide margin. The performance gap persists across all P@k levels, underscoring TIGER's ability to maintain reliable retrieval under the most difficult conditions. Crucially, TIGER achieves a mean rank of **219.8**, which, although still larger than in easier splits, is significantly lower than the 500+ ranks of all baselines. This confirms TIGER's strength in reducing the search depth required to identify correct enzyme matches even under severe distribution shifts.

# G OVERALL SUMMARY ACROSS SPLITS

The comprehensive experiments across the three evaluation splitsn(time-based, enzyme similarity-based, and reaction similarity-based) collectively demonstrate the robustness and generalizability of TIGER. Several consistent observations can be drawn from the results.

**Superior Top-1 Accuracy.** Across all splits and both retrieval directions, TIGER delivers the highest Hit@1 and P@1 scores, often by large margins. For example, in the time-based split TIGER attains 0.5810 Hit@1 for enzyme-to-reaction retrieval, compared to 0.3911 for the strongest baseline (Bi-RNN[2]). In the more challenging enzyme similarity-based split, this advantage becomes

Table 14: Reaction to Enzyme Retrieval Performance (P@k and Mean Rank) on Reaction Similarity-based Split.

| Method | P@1 | P@2 | P@3 | P@4 | P@5 | P@10 | P@20 | Mean Rank |
|---|---|---|---|---|---|---|---|---|
| ReactZyme[1] | 0.0924 | 0.0832 | 0.0812 | 0.0762 | 0.0721 | 0.0694 | 0.0591 | 548.3340 |
| ReactZyme[2] | 0.1347 | 0.1269 | 0.1218 | 0.1095 | 0.1083 | 0.0902 | 0.0749 | 529.4258 |
| ReactZyme[3] | 0.1347 | 0.1269 | 0.1218 | 0.1095 | 0.1083 | 0.0902 | 0.0749 | 529.4258 |
| Fingerprint | 0.1143 | 0.1047 | 0.1015 | 0.0987 | 0.0935 | 0.0851 | 0.0706 | 535.6742 |
| GNN[1] | 0.1244 | 0.1231 | 0.1166 | 0.1114 | 0.1114 | 0.0946 | 0.0775 | 559.1225 |
| GNN[3] | 0.0917 | 0.0861 | 0.0819 | 0.0783 | 0.0768 | 0.0716 | 0.0608 | 552.4546 |
| Bi-RNN[1] | 0.1244 | 0.1231 | 0.1166 | 0.1101 | 0.1036 | 0.0951 | 0.0790 | 545.8586 |
| Bi-RNN[2] | 0.1710 | 0.1464 | 0.1382 | 0.1367 | 0.1290 | 0.1145 | 0.0870 | 529.3677 |
| CLIPZyme[1] | 0.1298 | 0.1225 | 0.1044 | 0.0921 | 0.0866 | 0.0830 | 0.0741 | 526.4793 |
| CLIPZyme[2] | 0.1457 | 0.1291 | 0.1233 | 0.1156 | 0.1135 | 0.1001 | 0.0783 | 501.2071 |
| TIGER (Ours) | **0.4305** | **0.3756** | **0.3316** | **0.3069** | **0.2854** | **0.2269** | **0.1680** | **219.7977** |

even more pronounced, with TIGER reaching 0.9308 Hit@1 against 0.8151 for Bi-RNN[2]. In the reaction similarity-based split,the most difficult scenario,TIGER still secures 0.4155 Hit@1, nearly quadrupling the performance of prior methods. These results highlight TIGER's strength in ranking the correct match at the very top.

**Ranking Efficiency.** TIGER consistently achieves much lower mean ranks than all baselines, showing that it brings correct matches substantially closer to the top of the ranked lists. In enzyme-to-reaction retrieval under the enzyme similarity-based split, TIGER records a mean rank of only 1.58, while the best baseline remains at 2.71. Even in the most challenging reaction similarity-based split, TIGER reduces the mean rank to 23.2, compared with 35-500 for baselines. This efficiency is critical for real-world retrieval systems, where narrowing the search space is essential for practical usability.

**Robustness Across Retrieval Directions.** TIGER exhibits balanced improvements in both retrieval directions. While existing methods often show asymmetric performance, performing relatively better in enzyme-to-reaction but weaker in reaction-to-enzyme retrieval, TIGER maintains strong results in both cases. This symmetry indicates that TIGER captures a shared latent representation that generalizes effectively across modalities.

**Generalization Under Distribution Shifts.** Most importantly, TIGER's gains are preserved under increasing levels of distributional difficulty. In the time-based split, TIGER demonstrates strong forward-looking generalization; in the enzyme similarity-based split, it effectively handles test enzymes with little sequence homology to training examples; and in the reaction similarity-based split, it generalizes to novel reaction types where structural overlap is minimal. Across all three, TIGER outperforms baselines not only in absolute accuracy but also in robustness and efficiency.

**Conclusion.** Together, these results establish TIGER as a state-of-the-art framework for enzyme-reaction retrieval. It consistently surpasses strong baselines, achieves substantial improvements across diverse metrics, and demonstrates resilience under severe distribution shifts. The consistent superiority across all three splits confirms TIGER's capacity to generalize beyond simple sequence or reaction similarity, enabling reliable and efficient retrieval in realistic biochemical discovery scenarios.

# H LIMITATIONS

Our current framework has two main limitations. First, the processing of textual descriptions is relatively coarse-grained, which may overlook fine-grained catalytic details important for retrieval. Second, the evaluation of text quality remains implicit, and a more explicit assessment framework

could further strengthen the reliability of text-informed representations. We leave these aspects for future work.

