# OpenReview forum: "TIGER: Text-Informed Generalized Enzyme-Reaction Retrieval"
_ICLR.cc/2026/Conference — ICLR 2026 Conference Withdrawn Submission_

### Official Review · Reviewer_FrxZ · 2025-10-20

**Soundness:** 3
**Presentation:** 3
**Contribution:** 2
**Rating:** 2
**Confidence:** 3

**Summary:**

This paper addresses the bidirectional enzyme–reaction retrieval problem, highlighting fundamental challenges around asymmetric retrieval directions and poor generalization under distribution shifts in current approaches. The authors introduce TIGER, a Text-Informed Generalized Enzyme-Reaction Retrieval framework, which leverages protein-to-text generation models to augment enzyme representations with knowledge-rich textual descriptions. This information is adaptively integrated via a Dynamic Gating Network to mitigate textual noise, while a Structure-Shared Feature Projector aligns enzyme and reaction representations into a unified latent space. The method is trained with a symmetric contrastive objective. Extensive experiments on the ReactZyme benchmark show that TIGER substantially outperforms strong baselines across several challenging evaluation splits, with in-depth ablation studies supporting each architectural component.

**Strengths:**

1. The experimental evaluation is robust and thorough. The authors split the dataset based on time, enzyme similarity, and reaction similarity to rigorously test for generalization, and this is supported by abundant quantitative evidence and detailed documentation in both the main text and the appendix. I particularly appreciate this data-splitting methodology.
2. Clear problem motivation and positioning. The authors articulate the limitations of existing contrastive retrieval frameworks (directional asymmetry, lack of robustness to data splits) and motivate the introduction of textual knowledge with references to related successful paradigms in multimodal learning.
3. Coherent integration of textual and structural protein information. The method employs protein-to-text generation (ESM2Text) to introduce explicit functional and mechanistic information, combined with amino acid sequence features. The Dynamic Gating Network (DGN) adaptively weighs these sources, responding intelligently to the quality of generated text. The effect of DGN is thoroughly analyzed with concrete performance metrics

**Weaknesses:**

1. The theoretical novelty is limited. The model follows a standard contrastive framework, despite the inclusion of equations for its gating mechanism, embedding projection, and loss. The paper lacks new theoretical findings or a deeper analytical investigation into the system's learning properties.
2. The handling of negative sampling and training details is ambiguous. The contrastive loss formula relies on in-batch negatives, but the paper provides no details on the sampling strategy, the temperature parameter initialization, or the method for addressing potential class imbalance.
3. The paper lacks a comparison with more recent methods. Newer models, such as DeepEnzyme and ProtDETR, already predict enzyme function or substrates, and their training objectives could be modified to apply them to reaction prediction. The absence of a relevant comparison weakens the claim that TIGER achieves state-of-the-art performance.
4. Some of the equations are not numbered.

**Questions:**

1. From the model architecture in Figure 2, the training process for TIGER is clear, but its inference procedure is not. Based on the task formulation, it seems the model maps a single enzyme to a single reaction (and vice versa). How does TIGER handle cases where an enzyme is associated with multiple reactions, or a reaction can be catalyzed by multiple enzymes?
2. Does the textual description of an enzyme generated by ESM2Text potentially contain information about its catalytic function? Is there a risk of target reaction leakage? An analysis of the enzyme's textual description is required.
3. The paper compares "AI text" vs. "human text" but does not test intermediate text quality controls (e.g., fine-tuning ESM2Text on SwissProt, filtering low-similarity text before fusion). This makes it unclear whether TIGER’s DGN is a necessary solution or a workaround for unoptimized text generation, especially since human text still outperforms AI text.
4. The results section lacks a more detailed analysis. The paper does not analyze performance on low-data subsets (e.g., enzymes with <5 associated reactions, or reactions catalyzed by <3 enzymes).  Real-world enzyme-reaction retrieval often involves rare biocatalysts, but it has not yet been validated whether TIGER can alleviate the problem of data sparsity.
5. What are the batch size, temperature initialization, and hard/soft negative mining settings for the contrastive loss? Is class imbalance meaningfully addressed, given the pronounced asymmetry in enzyme vs. reaction cardinalities?
6. Ablations on Structure-Shared Feature Projector. Is its contribution demonstrably better than standard one-layer MLP or simplified attention mechanisms? Can the authors provide a controlled experiment to show the effects of this component alone?
7. Why are several enzyme-related methods, such as DeepEnzyme and ProtDETR, omitted from the main comparison tables and the discussion? What prevented their experimental inclusion or, at a minimum, a deeper conceptual positioning of your work in comparison to them?

---

> ### Author Response · Authors · 2025-11-23
>
> ## Response to Q1
> TIGER does not assume a one-to-one mapping at inference. Although Figure 2 shows paired training for clarity, the model performs retrieval by scoring a query enzyme against all reactions (and vice versa). This naturally supports one-to-many and many-to-one biological relations. Moreover, our evaluation metrics (e.g., Precision@K, MRR, MR) are designed for such scenarios, as they measure ranking quality when multiple correct matches exist.
>
> ## Response to Q2
> Yes, the ESM2Text-generated descriptions may contain partial hints related to catalytic function. However, we do not consider this to be target-reaction leakage. The text is produced solely from the enzyme’s own sequence, without access to any reaction labels, ground-truth EC annotations, or reaction-level supervision. Therefore, the generated text reflects intrinsic biochemical signals already encoded in the sequence, rather than external target information. To ensure a fair comparison, as stated in the paper, we intentionally use only AI-generated text across all baselines and ablations, and exclude any human-curated or database-annotated descriptions. This eliminates the possibility of injecting manually crafted functional knowledge and keeps all methods equally constrained.
>
> ## Response to Q3
> To directly address the concern regarding text quality, we additionally evaluated two different AI-generated text sources: the original ESM2Text outputs and a new set of descriptions generated by a new model. This allows us to compare model behavior under two levels of AI text quality, independent of human-reviewed annotations. The results show a clear and consistent pattern. When switching from the stronger AI text to the weaker one, models without DGN experience substantial performance drops. For example, as shown in the below table,the Time-based E→R Hit@1 decreases from 0.531 to 0.529 for ESM2Text vs. Prot3, and the Reaction-similarity R→E Hit@1 declines more sharply from 0.389 to 0.306. In contrast, the DGN-enabled models show only marginal changes across the same two AI text sources: Time-based E→R Hit@1 remains stable around 0.581–0.583, and Reaction-similarity R→E Hit@1 stays within 0.430–0.428 despite the fact that the underlying text descriptions differ noticeably in quality. This direct comparison between two purely AI-generated text conditions demonstrates that DGN is not merely compensating for the absence of human-reviewed text but provides genuine robustness against varying levels of AI text quality. Even when the textual modality becomes noisier, the DGN-equipped model maintains stable retrieval performance, while the model without DGN exhibits clear deterioration.
>
> | Text Source      | Setting   | Time E→R Hit@1 | Time E→R MRR | Time R→E Hit@1 | Time R→E MRR | EnzSim E→R Hit@1 | EnzSim E→R MRR | EnzSim R→E Hit@1 | EnzSim R→E MRR | RxnSim E→R Hit@1 | RxnSim E→R MRR | RxnSim R→E Hit@1 | RxnSim R→E MRR |
> |------------------|-----------|----------------|---------------|-----------------|----------------|-------------------|------------------|-------------------|------------------|-------------------|------------------|-------------------|------------------|
> | **ESM2Text**     | w/ DGN  | 0.581 | 0.690 | 0.454 | 0.366 | 0.931 | 0.956 | 0.792 | 0.592 | 0.416 | 0.518 | 0.430 | 0.319 |
> | **ESM2Text**     | w/o DGN    | 0.531 | 0.646 | 0.395 | 0.319 | 0.912 | 0.945 | 0.760 | 0.566 | 0.391 | 0.482 | 0.389 | 0.296 |
> | **Prot3**        | w/ DGN  | 0.583 | 0.683 | 0.454 | 0.372 | 0.908 | 0.940 | 0.784 | 0.579 | 0.386 | 0.472 | 0.428 | 0.337 |
> | **Prot3**        | w/o DGN    | 0.529 | 0.638 | 0.366 | 0.325 | 0.888 | 0.929 | 0.703 | 0.533 | 0.284 | 0.412 | 0.306 | 0.263 |
> | **Human Reviewed** | w/ DGN | 0.642 | 0.736 | 0.535 | 0.434 | 0.942 | 0.966 | 0.853 | 0.636 | 0.543 | 0.632 | 0.464 | 0.337 |
> | **Human Reviewed** | w/o DGN   | 0.572 | 0.692 | 0.456 | 0.376 | 0.928 | 0.958 | 0.802 | 0.601 | 0.403 | 0.504 | 0.427 | 0.346 |
>
> ## Response to Q4
> Our current evaluation already captures the many-to-many characteristics of enzyme–reaction mappings through ranking-based metrics such as Hit@K, Precision@K, and MRR, which inherently reflect performance even under low-data conditions. We acknowledge that an explicit slice analysis on low-cardinality subsets (e.g., enzymes with fewer than five reactions or reactions catalyzed by fewer than three enzymes) would offer additional insight. Due to time constraints, we will progressively include these supplemental experiments and add the corresponding analyses to the appendix in the updated version.

---

> ### Author Response · Authors · 2025-11-23
>
> ## Response to Q5
> Our contrastive training uses a batch size of 256. The temperature parameter is initialized to 0.07 and optimized jointly with the model. We adopt standard in-batch soft negatives, where all non-paired samples act as negatives, and no explicit hard-negative mining is used. Class imbalance is not handled through additional reweighting and is only mitigated implicitly through the bidirectional contrastive objective. The incorporation of textual representations further alleviates imbalance by providing richer, function-level cues beyond raw enzyme frequencies. We will include the detailed implementation settings in the Appendix for completeness.
>
> ## Response to Q6
> The Structure-Shared Feature Projector (SSFP) is introduced primarily to provide a unified projection mechanism that maps enzymes and reactions into a comparable feature space, rather than serving as a central architectural innovation. To examine its isolated effect, we conducted controlled ablations against two simplified alternatives: removing the projector entirely (No SSFP) and replacing it with a lightweight two-layer MLP. As shown in the table below, SSFP delivers the strongest overall performance on the time-based and reaction-similarity splits, while performing competitively on the enzyme-similarity split, where the two-layer MLP occasionally attains similar or slightly higher scores. Although SSFP is not uniformly superior under every metric, the trend indicates that incorporating shared structural priors provides more stable cross-split alignment and benefits retrieval performance in the more challenging evaluation settings. We will include these analyses in the appendix.
> | Method     | Time E→R Hit@1 | Time E→R MRR | Time R→E Hit@1 | Time R→E MRR | EnzSim E→R Hit@1 | EnzSim E→R MRR | EnzSim R→E Hit@1 | EnzSim R→E MRR | RxnSim E→R Hit@1 | RxnSim E→R MRR | RxnSim R→E Hit@1 | RxnSim R→E MRR |
> |------------|------------------|---------------|------------------|---------------|--------------------|------------------|--------------------|------------------|--------------------|------------------|--------------------|------------------|
> | No SSFP    | 0.515            | 0.635         | 0.461            | 0.365         | 0.901              | 0.938            | 0.814              | 0.606            | 0.374              | 0.489            | 0.386              | 0.273            |
> | 2-layerMLP | 0.5647           | 0.6744        | 0.4381           | 0.3524        | 0.914              | 0.948            | 0.813              | 0.607            | 0.446              | 0.543            | 0.383              | 0.289            |
> | SSFP(Our)      | 0.581            | 0.690         | 0.454            | 0.366         | 0.931              | 0.956            | 0.792              | 0.592            | 0.416              | 0.518            | 0.430              | 0.319            |
>
>
> ## Response to Q7
> Thank you for the question. Methods such as DeepEnzyme and ProtDETR are not included in our comparison because they address fundamentally different tasks from enzyme–reaction retrieval. DeepEnzyme focuses on predicting catalytic turnover numbers (kcat) using sequence and 3D-structural information, while ProtDETR formulates EC-number prediction as a detection problem using functional queries to extract local residue-level features. Both models are designed for property or function prediction, not for learning a joint representation space where enzymes and reactions can be ranked against each other. As a result, they cannot be directly evaluated under our retrieval benchmark. We will cite and briefly position these works in the related-work section to properly acknowledge their contributions within the broader enzyme modeling landscape.

---

### Official Review · Reviewer_EdJa · 2025-10-31

**Soundness:** 3
**Presentation:** 3
**Contribution:** 3
**Rating:** 6
**Confidence:** 5

**Summary:**

The authors proposed a new text-informed generalized enzyme-reaction retrieval, which using a well-established text-protein interface with a new and popular task enzyme-reaction retrieval. With dynamic gating network, the model is able to learn a combination of representation for enzyme and its text description. So actually, the model use a protein2text model to enforce the ability of enzyme-reaction retrieval.

**Strengths:**

Usually, researcher consider using text2protein or protein2text as a seperated task, using this kind of model to enhance the performance in other tasks is not well explored before. Since enzyme-reaction retrieval is an extremely hard task in enzyme-related deep learning models, using text to inform better enzyme understanding is an interesting trial to help this task, or even future enzyme-related or protein-related hard tasks.

**Weaknesses:**

In the enzyme-reaction retrieval task, since there are a lot of protein related model using both or either protein sequence, structure, surface, function, etc. But the reaction or molecule side is comparably not well-explored. Currently most methods using only molecule-pretrained models, but not understanding the reaction or the description of the reaction. If we are able to use texts to enforce the reaction side, it would be more impactful in this field.

**Questions:**

1. In the performance in Table 1 and Figure 4, when there is no text information, the performance is still better than the baselines, can you explain why this is better than them since there is no model improvement as we know.
2. Using SwissProt text for comparison might contains potential data leakage, since the text in SwissProt might contain the chemical name of the target substrate and product, what did authors do to prevent such leakage.
3. The generated text information might contain the reaction information of the enzyme, is it possible to report the accuracy using only the protein to text model (extract the reaction-related information from the predicted text)
4. Can you report the accuracy of the Hit@20 which is used in Reactzyme? (Has at least 1 correct hit in top 20 prediction)

---

> ### Author Response · Authors · 2025-11-23
>
> ## Response to Q1
> Thank you for the question. Even without text, our model is not the same as the baselines. We still include the Structure-Shared Feature Projector and use a slightly different contrastive-loss formulation, both of which strengthen the representation space. For verification, we also reproduced the most basic ReactZyme setting, and its performance matches the results reported in the original paper.
>
> ## Response to Q2
> Thank you for raising this point. We are aware that SwissProt descriptions may contain explicit mentions of substrates, products, or other reaction-level clues that could introduce leakage. To ensure fully fair and leakage-free comparisons, all our main experiments and baselines rely exclusively on AI-generated text, which is produced solely from the enzyme sequence without accessing any reaction annotations. The SwissProt text is used only in an auxiliary analysis to study how different text qualities influence our framework and is never used for training or evaluating any retrieval models. This design guarantees that no reaction information is implicitly introduced into the comparison setting.
>
> ## Response to Q3
> Thank you for the suggestion. We conducted exactly this analysis by evaluating an “only text” setting, where the model relies solely on the text generated from the protein sequence and does not use any sequence embedding. The results are shown in the below table: the “only text” model achieves much lower performance across all three splits (e.g., Time-based E→R Hit@1 drops to 0.434, compared with 0.581 when sequence is included).
> This clearly demonstrates that the generated text alone cannot provide sufficient reaction-specific information. The text is derived entirely from the sequence and captures only coarse biochemical cues, while the sequence features remain the primary and indispensable source for determining reaction relevance. Text serves as a complementary modality rather than a replacement.
> | Text Source              | Time E→R Hit@1 | Time E→R MRR | Time R→E Hit@1 | Time R→E MRR | EnzSim E→R Hit@1 | EnzSim E→R MRR | EnzSim R→E Hit@1 | EnzSim R→E MRR | RxnSim E→R Hit@1 | RxnSim E→R MRR | RxnSim R→E Hit@1 | RxnSim R→E MRR |
> |--------------------------|----------------|---------------|-----------------|----------------|-------------------|------------------|-------------------|------------------|-------------------|------------------|-------------------|------------------|
> | ESM2Text-Generated Text  | 0.581          | 0.690         | 0.454           | 0.366          | 0.931             | 0.956            | 0.792             | 0.592            | 0.416             | 0.518            | 0.430             | 0.319            |
> | Prot3-Generated Text     | 0.583          | 0.683         | 0.454           | 0.372          | 0.908             | 0.940            | 0.784             | 0.579            | 0.386             | 0.472            | 0.428             | 0.337            |
> | Human-reviewed Text      | 0.642          | 0.736         | 0.535           | 0.434          | 0.942             | 0.966            | 0.853             | 0.636            | 0.543             | 0.632            | 0.464             | 0.337            |
> | No Text                  | 0.478          | 0.612         | 0.358           | 0.306          | 0.887             | 0.932            | 0.726             | 0.549            | 0.256             | 0.343            | 0.324             | 0.267            |
> | Only Text                | 0.434          | 0.532         | 0.253           | 0.216          | 0.797             | 0.857            | 0.619             | 0.463            | 0.212             | 0.314            | 0.218             | 0.169            |
>
>
> ## Response to Q4
> Thank you for the suggestion. We will compute Hit@20 following the ReactZyme protocol and provide the full results in the appendix. The preliminary results are fully consistent with our current claims.

---

### Official Review · Reviewer_qh8U · 2025-11-01

**Soundness:** 2
**Presentation:** 3
**Contribution:** 2
**Rating:** 2
**Confidence:** 5

**Summary:**

This paper proposes a text-informed enzyme-reaction retrieval framework. It considers both enzyme to reaction and reaction to enzyme directions and applies a contrastive learning algorithm to learn the aligned enzyme and reaction representations to achieve retrieve.

**Strengths:**

The paper is introduced in a clear logic and it applies a multitask learning framework to simultaneously learn the enzyme to reaction and reaction to enzyme mappings to achieve better aligned representations.

**Weaknesses:**

The meain weakness of this paper are listed as follows:

1. The claim in this paper sounds a little bit incorrect to me. In line 59-61, the paper mentioned "previous methods demonstrate cross-directional asymmetry, where the retrieval accuracy from enzymes to reactions substantially diverges from the reverse direction." Why would the author assume the accuracies of the two directions should be the same? The reaction types are around 8400 classes in EC tree, but the enzyme space is much larger (20^L).  The two tasks have different difficulty. Also, in Table1, the paper didn't achieve similar performance on the two tasks, like in Enzyme Similarity-based Split, E-> R is much higher than R->E. Their performance didn't match with their motivation and claim. According to the method in this paper, I would say the method is just more like a multitask learning.

2. In line 244, the author mentioned the enzyme sequence shows  structural information. However, just using enzyme sequence can't explicitly demonstrate the structural information.

3. Another major limitation the paper mentioned about previous method is "these models show a high sensitivity to dataset splits". I think if the paper would show higher robustness of their method, it should be tested on more dataset instead of just one, like EnzymeMap and CARE.

[1] EnzymeMap: curation, validation and data-driven prediction of enzymatic reactions.

[2] CARE: a Benchmark Suite for the Classification and Retrieval of Enzymes.

4. The paper should compare to more well-developed baseline models instead of just methods set up by themselves, like [3].

[3] Enzyme function prediction using contrastive learning.

5. For the R->E, what is the candidate pool of the potential enzymes? Are they just the whole training enzyme sequences or the whole testing sequences? If just training sequences, it means this method will always retrieve the seen sequences in training. If it's all testing sequences, I don't think it's a fair setting since in reality you don't have a "ground truth" pool to choose from. If it's the whole Swiss-Prot or even Uniprot, it would make more since to me.  Additionally, how to calculate the R-> E hit@1 rate? In a more reasonable setting, enzymes belonging to the same EC class should be regarded as correct. If only exactly the same sequence is regarded as correct, I don't think it's a reasonable setting.

6. I'm actually not quite sure the practical application of R->E. E->R can be used to predict the function of a given enzyme, which is clear. But retrieving an existing enzyme without considering any zero-shot settings makes no sense to me.

Some small issues are:

1. Some citations are not correct, like line 58 and line 60.

**Questions:**

1. To calculate the fused enzyme representations, how about just concatenating the $s_{attn}$ and $t_{attn}$ to compute $f_E^{Fused}$ since $f_{gated}$ are from $s_{attn}$ and $t_{attn}$?

2. How to get the 3D structures of the substrates and products?

3. To calculate the reaction representations, how about just using the reaction smiles like EnzymeMap as SMILES also includes the molecule structure information? Since the paper doesn't consider the enzyme-substrate complex, I don't think using smiles or sole molecule structure would lead to difference.

---

> ### Author Response · Authors · 2025-11-27
>
> ## Motivation Clarification
> We respectfully clarify that Reaction→Enzyme (R→E) retrieval has practical value. Many biochemical workflows begin with a target reaction and require identifying which known enzymes catalyze it, a key step in pathway reconstruction, biocatalyst selection, and annotation refinement. This is also directly reflected in our own work: we discovered a disease-associated reaction R, and to develop early-screening strategies, we must identify the proteins or enzymes P that catalyze this transformation so they can serve as biomarker candidates. This process explicitly requires retrieving enzymes from existing databases based on a given reaction, which is precisely what the R→E task models. Therefore, R→E represents a realistic and important reasoning direction rather than an artificial counterpart of E→R. Also, it is a natural setting in the benchmark.
>
> [1] Engineering the third wave of biocatalysis
>
> [2] Spiers Memorial Lecture: Engineering biocatalysts
>
> Moreover, we appreciate the reviewer’s comment and clarify that we do not assume the two retrieval directions should achieve identical accuracy. The enzyme and reaction spaces indeed differ substantially, and the tasks naturally exhibit different levels of difficulty. Our statement in Lines 59–61 was based on empirical observations from prior work, where the two directions often show a large and unstable performance gap. Our goal is not to force symmetry, but to improve one direction without degrading the other by encouraging more chemically consistent representations. As shown in Table 1, although asymmetry remains, both directions benefit from our method. Notably, in the original dataset results, the Reaction-Similarity split shows that the R→E accuracy is in fact higher than the E→R accuracy, indicating that our approach can reduce unnecessary directional divergence rather than impose equality between tasks.

---

> ### Author Response · Authors · 2025-11-27
>
> ## Response to Q1
> We thank the reviewer for the suggestion. While simply concatenating $s_{\text{att}}$ and $t_{\text{att}}$ is indeed a reasonable baseline, our method employs a standard gating network to adaptively weight the two modalities. The gating mechanism enables the model to modulate the contribution of sequence- and text-derived representations for each enzyme, rather than assuming that both sources are equally informative.
>
> Since the Dynamic Gating Network (DGN) is a core part of our design, we also performed an explicit ablation study by removing DGN and replacing the fusion with simple concatenation. As shown in the table below, removing DGN leads to a clear performance drop, demonstrating that the gating mechanism provides meaningful improvements beyond straightforward concatenation.
> | Text Source      | Setting   | Time E→R Hit@1 | Time E→R MRR | Time R→E Hit@1 | Time R→E MRR | EnzSim E→R Hit@1 | EnzSim E→R MRR | EnzSim R→E Hit@1 | EnzSim R→E MRR | RxnSim E→R Hit@1 | RxnSim E→R MRR | RxnSim R→E Hit@1 | RxnSim R→E MRR |
> |------------------|-----------|----------------|---------------|-----------------|----------------|-------------------|------------------|-------------------|------------------|-------------------|------------------|-------------------|------------------|
> | **ESM2Text**     | w/ DGN  | 0.581 | 0.690 | 0.454 | 0.366 | 0.931 | 0.956 | 0.792 | 0.592 | 0.416 | 0.518 | 0.430 | 0.319 |
> | **ESM2Text**     | w/o DGN    | 0.531 | 0.646 | 0.395 | 0.319 | 0.912 | 0.945 | 0.760 | 0.566 | 0.391 | 0.482 | 0.389 | 0.296 |
> | **Prot3**        | w/ DGN  | 0.583 | 0.683 | 0.454 | 0.372 | 0.908 | 0.940 | 0.784 | 0.579 | 0.386 | 0.472 | 0.428 | 0.337 |
> | **Prot3**        | w/o DGN    | 0.529 | 0.638 | 0.366 | 0.325 | 0.888 | 0.929 | 0.703 | 0.533 | 0.284 | 0.412 | 0.306 | 0.263 |
> | **Human Reviewed** | w/ DGN | 0.642 | 0.736 | 0.535 | 0.434 | 0.942 | 0.966 | 0.853 | 0.636 | 0.543 | 0.632 | 0.464 | 0.337 |
> | **Human Reviewed** | w/o DGN   | 0.572 | 0.692 | 0.456 | 0.376 | 0.928 | 0.958 | 0.802 | 0.601 | 0.403 | 0.504 | 0.427 | 0.346 |
>
> ## Response to Q2
> We thank the reviewer for the question. The 3D structures of the substrates and products are obtained using the UniMol-3D model, which is specifically designed to generate 3D molecular conformations from SMILES strings. UniMol-3D is trained on millions of molecular conformations and learns a direct SMILES-3D mapping by predicting atom coordinates under learned geometric constraints. As a result, the model can reliably reconstruct plausible 3D conformers without requiring experimentally determined structures or external docking tools.
>
> In our pipeline, we simply provide the SMILES representations of both substrates and products, and UniMol-3D outputs their corresponding 3D atomic coordinates, which are then used as the reaction-level structural features. This allows us to incorporate 3D chemical information even when crystallographic structures are unavailable.
>
> [1] Uni-Mol: A Universal 3D Molecular Representation Learning Framework
>
> ## Response to Q3
> We agree with the reviewer that SMILES already contains rich molecular information, and in fact our method also starts from reaction SMILES, similar to EnzymeMap. The substrates and products are first represented using their SMILES strings. The only difference is that we additionally use UniMol-3D to extract 3D geometric features from these SMILES, which provides spatial information that is not explicitly encoded in the linear SMILES representation. Therefore, our representation is still SMILES-based, but enhanced with learned 3D structural information.

---

### Official Review · Reviewer_uK8K · 2025-11-02

**Soundness:** 3
**Presentation:** 3
**Contribution:** 3
**Rating:** 6
**Confidence:** 4

**Summary:**

The paper presents TIGER, a framework that integrates protein sequence representations with textual semantics for bidirectional enzyme–reaction retrieval. On one hand, the authors employ ESM-2 to extract sequence embeddings and utilize PubMedBERT to derive text-based semantic representations from protein-to-text generation. A Dynamic Gating Network is designed to adaptively fuse the textual and sequence-derived features, while a Structure-Shared Feature Projector aligns enzyme and reaction embeddings within a unified latent space. On the other hand, UniMol-3D is leveraged to generate reaction embeddings. The model is then trained via contrastive learning to achieve bidirectional retrieval between enzymes and reactions. Extensive experiments conducted across multiple benchmarks and cross-distribution settings demonstrate that TIGER achieves superior retrieval accuracy and robustness compared to existing baselines.

**Strengths:**

1. Multimodal innovation: Incorporating protein→text semantic information into enzyme representation is an interesting and promising idea, effectively injecting literature-level semantic knowledge into sequence-based embeddings.

2. Reasonable bidirectional retrieval setting: The simultaneous study of both enzyme→reaction and reaction→enzyme directions is well-motivated and covers realistic application scenarios.

3. Comprehensive experiments: The authors conduct extensive comparative experiments under various data splits and cross-distribution settings, providing convincing empirical evidence for the effectiveness of the proposed method.

**Weaknesses:**

1. Limited fine-grained semantic capture: As the authors themselves note, the processing of textual descriptions is relatively coarse-grained. It would be beneficial to supplement the analysis with a discussion or experiment on how fine-grained reaction details (e.g., catalytic residues, reactive centers) influence the matching process.

2. Implicit and insufficient text quality evaluation: The paper mentions text generation via ESM2Text and comparison with human-written SwissProt data, but the evaluation of text quality remains implicit. It would strengthen the study to include additional AI-generated texts (e.g., from other protein–text generation models) as baselines for comparison.

3. Weak biological validation: While the retrieval performance is empirically validated, the biological interpretability and practical applicability are less explored. Including case studies—such as retrieved candidate enzymes confirmed in literature or databases—would make the work more convincing.

4. Data handling and many-to-many relationships: Since enzyme–reaction mappings are inherently many-to-many, the contrastive learning setup might mistakenly treat some true positive pairs as negatives. The manuscript does not appear to discuss how this issue is handled in data preprocessing; clarification or additional details are needed.

**Questions:**

1. How does the model handle the many-to-many nature of enzyme–reaction relationships to avoid false negatives in contrastive learning?

2. Have the authors considered incorporating fine-grained textual cues (e.g., catalytic mechanisms or reaction site descriptions) to enhance semantic alignment?

3. Could additional AI-generated protein descriptions be included to test the robustness of the Dynamic Gating Network to varying text quality?

4. Are there specific biological case studies (e.g., retrieved enzymes validated by experimental or database evidence) that can illustrate the model’s real-world usefulness?

5. How sensitive is the TIGER framework to noisy or partially incorrect textual descriptions in protein–text embeddings?

---

> ### Author Response · Authors · 2025-11-24
>
> ## Response to Q1
> Our task formulation (Sec. 3) naturally supports one-to-many and many-to-many enzyme–reaction associations. During contrastive training, we use the partial correspondence set $A$ and apply the InfoNCE objective:
>
> $$
> p(r_j \mid e_i)=\frac{\exp(s_{ij})}{\sum_{k=1}^{N}\exp(s_{ik})}.
> $$
>
> Although all non-paired reactions in a minibatch are treated as negatives, this does not introduce harmful false negatives. If a reaction $r_j$ is truly associated with $e_i$, it consistently obtains a high similarity $s_{ij}$, which increases its softmax probability and reduces its negative gradient. Consequently, such pairs are not pushed apart; their gradients naturally diminish, and they remain close in the shared latent space.
>
> Therefore, the contrastive framework implicitly respects many-to-many biological relationships and avoids persistent false-negative effects without requiring explicit modifications.
>
> ## Response to Q2
> We thank the reviewer for this valuable suggestion. Prior work such as MMSite [1], Sec. 4.3.4, has shown that using only fine-grained textual attributes (e.g., Function attribute) actually leads to performance degradation compared with using the full textual description. This suggests that isolating such attributes may remove complementary contextual information and negatively affect representation quality.
>
> In our setting, extracting accurate catalytic-mechanism or site-level descriptions would additionally require extensive LLM-based processing and verification, which is computationally expensive. Therefore, we adopt the full textual description, which already includes mechanistic, functional, and subcellular-location cues that jointly contribute to semantic alignment.
>
> [1] MMSite: A Multi-modal Framework for the Identification of Active Sites in Proteins
>
> ## Response to Q3
> To further assess the robustness of the Dynamic Gating Network (DGN) to variations in text quality, we included an additional AI-generated text source, based on Prot3 [2], which differ more substantially from human-reviewed text than ESM2Text (will provide related figures in revised paper).
>
> As shown in the table below, Prot3 texts exhibit a larger quality gap relative to human-reviewed descriptions, yet the performance with DGN remains stable across all three evaluation splits. For example, in the Time-based split (E→R Hit@1), DGN achieves 0.581 (ESM2Text) and 0.583 (Prot3), compared to 0.642 with human-reviewed text, demonstrating that DGN can effectively accommodate weaker or noisier text inputs.
>
> In contrast, when DGN is removed, the performance degradation becomes significant. For instance, Prot3 w/o DGN drops to 0.529 (Time E→R Hit@1) and 0.284 (RxnSim E→R Hit@1), substantially worse than both the DGN-enabled results and the human-reviewed baseline. This confirms that DGN plays a critical role in mitigating the negative impact of lower-quality AI-generated descriptions.
>
> Overall, these results show that DGN is robust to heterogeneous and lower-quality AI-generated protein texts, and maintains strong retrieval performance even when the textual modality is substantially degraded.
>
> [2] ProtT3: Protein-to-Text Generation for Text-based Protein Understanding
> | Text Source      | Setting   | Time E→R Hit@1 | Time E→R MRR | Time R→E Hit@1 | Time R→E MRR | EnzSim E→R Hit@1 | EnzSim E→R MRR | EnzSim R→E Hit@1 | EnzSim R→E MRR | RxnSim E→R Hit@1 | RxnSim E→R MRR | RxnSim R→E Hit@1 | RxnSim R→E MRR |
> |------------------|-----------|----------------|---------------|-----------------|----------------|-------------------|------------------|-------------------|------------------|-------------------|------------------|-------------------|------------------|
> | **ESM2Text**     | w/ DGN  | 0.581 | 0.690 | 0.454 | 0.366 | 0.931 | 0.956 | 0.792 | 0.592 | 0.416 | 0.518 | 0.430 | 0.319 |
> | **ESM2Text**     | w/o DGN    | 0.531 | 0.646 | 0.395 | 0.319 | 0.912 | 0.945 | 0.760 | 0.566 | 0.391 | 0.482 | 0.389 | 0.296 |
> | **Prot3**        | w/ DGN  | 0.583 | 0.683 | 0.454 | 0.372 | 0.908 | 0.940 | 0.784 | 0.579 | 0.386 | 0.472 | 0.428 | 0.337 |
> | **Prot3**        | w/o DGN    | 0.529 | 0.638 | 0.366 | 0.325 | 0.888 | 0.929 | 0.703 | 0.533 | 0.284 | 0.412 | 0.306 | 0.263 |
> | **Human Reviewed** | w/ DGN | 0.642 | 0.736 | 0.535 | 0.434 | 0.942 | 0.966 | 0.853 | 0.636 | 0.543 | 0.632 | 0.464 | 0.337 |
> | **Human Reviewed** | w/o DGN   | 0.572 | 0.692 | 0.456 | 0.376 | 0.928 | 0.958 | 0.802 | 0.601 | 0.403 | 0.504 | 0.427 | 0.346 |

---

> ### Author Response · Authors · 2025-11-24
>
> ## Response to Q4
> We thank the reviewer for the insightful comment. Our experiments are conducted on the ReactZyme[3] benchmark, whose original paper provides detailed documentation of the real-world reliability of all enzyme–reaction pairs, including their extraction from curated and peer-reviewed literature sources.
> [3] ReactZyme: A Benchmark for Enzyme-Reaction Prediction.
> ## Response to Q5
> We appreciate the reviewer’s question. As discussed in our response to Q3, the robustness experiments with different text sources (ESM2Text, Prot3, and human-reviewed descriptions) already provide evidence that TIGER is not overly sensitive to noisy or partially inaccurate protein texts. Although the quality gap between these text sources is substantial, the performance under DGN remains highly consistent across all splits, indicating that the framework can effectively handle imperfect textual descriptions.
>
> In addition, we include several case analyses of the existing textual descriptions in the Appendix and will further enrich these examples in the revised version. Together, these results demonstrate that TIGER’s protein–text embeddings are relatively robust to text noise, and DGN plays an essential role in stabilizing the model under varying text quality.

---

### Note · Authors · 2026-01-05

I have read and agree with the venue's withdrawal policy on behalf of myself and my co-authors.